# ONE-STEP ESTIMATOR FOR PERMUTED SPARSE RECOVERY

## ABSTRACT

This paper considers the unlabeled sparse recovery under multiple measurements, i.e., $\mathbf{Y} = \mathbf{\Pi}^{\natural}\mathbf{X}\mathbf{B}^{\natural} + \mathbf{W}$, where $\mathbf{Y} \in \mathbb{R}^{n \times m}, \mathbf{\Pi}^{\natural} \in \mathbb{R}^{n \times n}, \mathbf{X} \in \mathbb{R}^{n \times p}, \mathbf{B}^{\natural} \in \mathbb{R}^{p \times m}, \mathbf{W} \in \mathbb{R}^{n \times m}$ represents the observations, missing (or incomplete) correspondence information, sensing matrix, sparse signals, and additive sensing noise, respectively. Different from the previous works on multiple measurements ($m > 1$) which all focus on the sufficient samples regime, namely, $n > p$, we consider a sparse matrix $\mathbf{B}^{\natural}$ and investigate the insufficient samples regime (i.e., $n \ll p$) for the first time. To begin with, we establish the lower bound on the sample number and *signal-to-noise ratio* (SNR) for the correct permutation recovery. Moreover, we present a simple yet effective estimator. Under mild conditions, we show that our estimator can restore the correct correspondence information with high probability. Numerical experiments are presented to corroborate our theoretical claims.

## 1 INTRODUCTION

In recent years, linear regression with permuted correspondence has received increasing attention due to its wide applications in the field of machine learning, signal processing, and statistics. Among all these applications, two most prominent examples are $(i)$ *linkage record*, which merges two datasets pertaining to the same objects into one comprehensive dataset; and $(ii)$ *data de-anonymization*, which infers the hidden labels of private data with public datasets. Apart from these two applications, other applications include *correspondence estimation* between pose and estimation in graphics; *time-domain sampling* in the presence of clock jitter; *multi-target tracking*; *unsupervised data alignment*, etc (Pananjady et al., 2018; Slawski & Ben-David, 2019; Slawski et al., 2020; Zhang et al., 2018).

In this paper, we consider the canonical model, i.e., a linear sensing relation with permuted labels:

$$\mathbf{Y} = \mathbf{\Pi}^{\natural}\mathbf{X}\mathbf{B}^{\natural} + \mathbf{W},$$

where $\mathbf{Y} \in \mathbb{R}^{n \times m}$ is the sensing result, $\mathbf{\Pi}^{\natural} \in \mathbb{R}^{n \times n}$ is an unknown permutation matrix, $\mathbf{X} \in \mathbb{R}^{n \times p}$ is the design (sensing) matrix, $\mathbf{B}^{\natural} \in \mathbb{R}^{p \times m}$ represents the sparse signals of interests, and $\mathbf{W} \in \mathbb{R}^{n \times m}$ denotes the additive noise. Assuming the signal $\mathbf{B}^{\natural}$ is a sparse signal, to put more specifically, each column of $\mathbf{B}^{\natural}$ is $k$-sparse, we would like to $(i)$ study the statistical limits of the permutation recovery under this scenario, e.g., the minimum sample number $n$ and *signal-to-noise ratio* (SNR); and $(ii)$ propose a practical estimator that can efficiently recover the permutation once the minimum requirements are met. To begin with, we briefly review the previous works.

**Related Works.** The study of permuted linear regression has a long history that can at least date back to DeGroot & Goel (1976; 1980); Goel (1975); Bai & Hsing (2005). Recent interests on this area start from Unnikrishnan et al. (2015). Focusing on the noiseless case $\mathbf{W} = \mathbf{0}$ with single measurement ($m = 1$), Unnikrishnan et al. (2015) establish the necessary condition $n \geq 2p$ for the permutation recovery if $\mathbf{B}^{\natural}$ is an arbitrary vector residing within the linear space $\mathbb{R}^p$. Later, Pananjady et al. (2018) extend the analysis to the noisy scenario. They showed the minimum SNR should be at least the order of $\Omega(n^c)$, where $c > 0$ is some positive constant. Numerical experiments suggest $c$ is within the region $[4, 5]$. Other works such as Hsu et al. (2017); Abid et al. (2017); Slawski & Ben-David (2019); Tsakiris et al. (2020); Haghighatshoar & Caire (2018) also focus on this regime and obtain the same answer. In Emiya et al. (2014), the setting with a sparse signal $\mathbf{B}^{\natural}$ is first studied. However, only empirical investigation is conducted without rigorous theoretical analysis. In the first work with theoretical analysis (Zhang & Li, 2021), both the statistical limits and practical estimators with almost optimal performance are presented for the permutation recovery. Peng et al. (2021) studies the

problem from the viewpoint of algebraic geometry. All existing works suggest that $\mathsf{SNR} = \Omega(n^c)$ is inevitable for the permutation reconstruction if only one measurement is conducted, namely, $m = 1$.

On the other hand, numerous works suggest multiple measurements, i.e., $m > 1$, can greatly reduce the SNR requirement, even to some positive constant. This line of research starts from Zhang et al. (2022), where the information theoretic lower bounds and the *maximum likelihood* (ML) estimator are investigated. Later, Zhang & Li (2020) study this problem from the viewpoint of non-convex optimization and propose an optimal estimator for the permutation recovery. Independently, Slawski et al. (2020) investigate this problem from the viewpoint of denoising. Putting parsimonious constraints on the number of permuted rows, they view $(\mathbf{I} - \mathbf{\Pi}^\natural)\mathbf{X}\mathbf{B}^\natural$ as sparse outliers and design the permutation recovery algorithm accordingly. These works focus on the sufficient samples regime, namely, $n = \Omega(p)$. In this paper, we focus on the insufficient samples regime. Assuming $\mathbf{B}^\natural$ to be sparse, we would like to show the correct permutation can be obtained with $n \ll p$ and $\mathsf{SNR} = O(1)$.

Our **contributions** are summarized as follows

- We study the lower bounds w.r.t the number $n$ and *signal-to-noise ratio* (SNR) for the correct reconstruction of both the permutation matrix $\mathbf{\Pi}^\natural$ and the signal $\mathbf{B}^\natural$. Assuming each column $\mathbf{B}^\natural_{:,\ell}$ $(1 \leq \ell \leq m)$ is $k$-sparse, we show that the sample number $n$ should be at least of the order $\Omega(k \log p)$; meanwhile the SNR should satisfy $\log \det(\mathbf{I} + \mathbf{B}^{\natural\top}\mathbf{B}^\natural/\sigma^2) > \log n + \frac{m \log \binom{p}{k}}{n}$.

- We propose a one-step estimator for the correspondence recovery, which consists of two sub-parts: one for $\mathbf{\Pi}^\natural$ and another for $\mathbf{B}^\natural$. By formulating the correspondence recovery as a *linear assignment problem* (LAP) (Kuhn, 1955; Bertsekas & Castañón, 1992; Burkard et al., 2012), the correct permutation matrix can be obtained when SNR is above certain positive constant.

On top of the above contributions, we would like to briefly mention our proof strategy, which is based on a tailored version of the **leave-one-out** technique. Compared with the previous works that adopt the leave-out-out technique (Chen et al., 2020; Sur et al., 2019; El Karoui, 2013; 2018; Cai et al., 2021), our construction method bas the following characteristics

- Our construction method is adaptive, which replaces multiple rows ranging from 2 to 4 depending on each permuted row. Meanwhile, previous works such as Chen et al. (2020); Sur et al. (2019); El Karoui (2013; 2018); Cai et al. (2021) replace a fixed number of rows (or columns).

- We not only leave out the rows, but also modify the thresholding operator operated on the perturbed samples $\widetilde{\mathbf{B}}_{(\cdots)}$ from $\mathsf{thres}(\cdot)$ to $(\cdot)_{\mathsf{imax}}$ (its definition is deferred to Subsection 4.2). This step is essential in controlling the approximation error, since otherwise the non-zero elements in matrices $\widehat{\mathbf{B}}$ $(\propto \mathbf{X}^\top \mathbf{Y})$ and $\widetilde{\mathbf{B}}_{(\cdot)}$ may not share the same position and the approximation error can be considerably large. A thorough understanding is deferred to the proof of Theorem 3.

**Notations.** Denote $c$, $c'$, $c_i$ as some positive constants, whose values are not necessarily the same even for those with the same notations. We denote $a \lesssim b$ if there exists some positive constants $c_0 > 0$ such that $a \leq c_0 b$. Similarly, we define $a \gtrsim b$ provided $a \geq c_0 b$ for some positive constant $c_0$. We write $a \asymp b$ when $a \lesssim b$ and $a \gtrsim b$ hold simultaneously.

For an arbitrary matrix $\mathbf{M}$, we denote $\mathbf{M}_{i,:}$ as its $i$th row, $\mathbf{M}_{:,i}$ as its $i$th column, and $\mathbf{M}_{ij}$ as its $(i, j)$th element. Its Frobenius norm is defined as $\|\mathbf{M}\|_{\mathsf{F}}$ while the operator norm is denoted as $\|\mathbf{M}\|_{\mathsf{OP}}$, whose definitions can be found in Section 2.3 of Golub & Loan (2013). In addition, we define its stable rank as $\mathsf{srank}(\cdot) \triangleq \|\cdot\|_{\mathsf{F}}^2/\|\cdot\|_{\mathsf{OP}}^2$ (Section 2.1.15 in Tropp (2015)) and its support set $\mathsf{supp}(\cdot)$ as $\{(i, j) : (\cdot)_{i,j} \neq 0\}$. The inner product between matrices is denoted as $\langle\!\langle \cdot, \cdot \rangle\!\rangle$ while the inner product between vectors is denoted as $\langle \cdot, \cdot \rangle$.

We define the set of all possible permutation matrices as $\mathcal{P}_n$, which is defined as $\{\mathbf{\Pi} \in \{0, 1\}^{n \times n} : \sum_{i=1}^n \mathbf{\Pi}_{ij} = 1, \sum_{j=1}^n \mathbf{\Pi}_{ij} = 1\}$. Associate with each permutation matrix $\mathbf{\Pi}$, we define the operator $\pi(\cdot)$ that transforms index $i$ to $\pi(i)$ under $\mathbf{\Pi}$. The Hamming distance $\mathsf{d}_{\mathsf{H}}(\mathbf{\Pi}_1, \mathbf{\Pi}_2)$ between two permutation matrices $\mathbf{\Pi}_1$ and $\mathbf{\Pi}_2$ is defined as $\mathsf{d}_{\mathsf{H}}(\mathbf{\Pi}_1, \mathbf{\Pi}_2) = \sum_{i=1}^n \mathbb{1}(\pi_1(i) \neq \pi_2(i))$. The SNR is defined as $\|\mathbf{B}^\natural\|_{\mathsf{F}}^2/(m \cdot \sigma^2)$.

## 2 PROBLEM FORMULATION

We start this section with a formal restatement of the considered problem reading as

$$\mathbf{Y} = \mathbf{\Pi}^\natural \mathbf{X} \mathbf{B}^\natural + \mathbf{W}, \tag{1}$$

where $\mathbf{\Pi}^\natural \in \mathcal{P}_n$ denotes the (fixed but unknown) permutation matrix, $\mathbf{X} \in \mathbb{R}^{n \times p}$ is the sensing (design) matrix with its entries being i.i.d. standard normal random variables, i.e., $\mathbf{X}_{ij} \sim \mathsf{N}(0, 1)$, [1] $\mathbf{B}^\natural \in \mathbb{R}^{p \times m}$ is a fixed sparse matrix awaiting to be reconstructed (corresponding to the sparse signal), and $\mathbf{W} \in \mathbb{R}^{n \times m}$ denote the noise with each entry $\mathbf{W}_{ij} \overset{\text{i.i.d}}{\sim} \mathsf{N}(0, \sigma^2)$. Here we put the separate sparse constraints on each column of $\mathbf{B}^\natural$, namely, $\|\mathbf{B}^\natural_{:,\ell}\|_0 \le k$ ($1 \le \ell \le m$). In addition, we denote $h$ as the number of permuted rows, or equivalently, the Hamming distance between the identify matrix $\mathbf{I}$ and the permutation matrix $\mathbf{\Pi}^\natural$, namely, $h \triangleq \mathsf{d}_\mathsf{H}(\mathbf{I}, \mathbf{\Pi}^\natural)$.

Our goal is to reconstruct both the permutation matrix $\mathbf{\Pi}^\natural$ and sparse signal $\mathbf{B}^\natural$ from (1). Note that we do not assume that different columns of $\mathbf{\Pi}^\natural$ share the same support set. Actually, we prefer each column to be with a different support set, since otherwise rank($\mathbf{B}^\natural$) will be bounded by $k$ and will bring extra difficulties to the permutation recovery. A detailed explanation is deferred to Section 4.

Before proceeding, we briefly review the prior art. In Unnikrishnan et al. (2015), where $\mathbf{B}^\natural$ can reside within the entire linear space $\mathbb{R}^p$, it is proved that $n \ge 2p$ is required for the correct permutation recovery. As a result, subsequent works such as Pananjady et al. (2018); Zhang et al. (2022); Slawski & Ben-David (2019); Slawski et al. (2020); Zhang & Li (2020) all focus on the sufficient sample regime, i.e., $n = \Omega(p)$. Only until Zhang & Li (2021), the insufficient sample regime, i.e., $n = o(p)$, receives its first theoretical investigation. Similar to our setting, they put sparsity assumption on $\mathbf{B}^\natural$ however focus on the single measurement scenario, namely, $m = 1$. They show that the minimum SNR for correct $(\mathbf{\Pi}^\natural, \text{supp}(\mathbf{B}^\natural))$ to be $\Omega(n^{c_1/\text{srank}(\mathbf{B}^\natural)} p^{c_2 p/n})$. In the following context, we will show that the minimum SNR can be significantly reduced, to put more specifically, some positive constant provided multiple measurements are made ($m \gg 1$).

## 3 INFORMATION THEORETIC LOWER BOUNDS

This section establishes the information theoretic lower bounds for the correct permutation recovery. Our goal is to ensure both $\mathbf{\Pi}^\natural$ and $\mathbf{B}^\natural$ can be reliably reconstructed. We investigate this problem from two perspectives: $(i)$ the sample number $n$ and $(ii)$ the minimum SNR.

### 3.1 THE MINIMUM SAMPLE NUMBER $n$

We obtain the minimum sample number $n$ such that sparse signal $\mathbf{B}^\natural$ can be reliably recovered with high probability. Here we consider the oracle situation where $\mathbf{\Pi}^\natural$ is given a prior. As each column in $\mathbf{B}^\natural$ does not necessarily share the same support set, we need to iteratively reconstruct each column $\mathbf{B}^\natural_{:,\ell}$, a $k$-sparse signal, from the corresponding readings $\mathbf{Y}_{:,\ell}$ ($1 \le \ell \le m$). With the classical result in Donoho (2006); Candes et al. (2006); Candès et al. (2006), we obtain the lower bound on $n$, namely, $n \gtrsim k \log p$. Naturally, we can expect this bound applies to the non-oracle situation as well, since reliable estimation of $\mathbf{B}^\natural$ is hopeless provided it is out of reach in the oracle situation.

### 3.2 THE MINIMUM SNR

Then we turn to the minimum SNR requirement for the correct permutation recovery. To begin with, we restate the prior art in Zhang et al. (2022).

**Theorem 1** (Theorem 1 in Zhang et al. (2022)). *Consider the oracle case where $\mathbf{B}^\natural$ is given a prior. Then there exists an integer $n_0$ such that for an arbitrary estimator $\widehat{\mathbf{\Pi}}$, we have*

$$\inf_{\widehat{\mathbf{\Pi}}} \sup_{\mathbf{\Pi}^\natural \in \mathcal{P}_n} \mathbb{P}_{\mathbf{X}, \mathbf{W}}(\widehat{\mathbf{\Pi}} \ne \mathbf{\Pi}^\natural) \ge \frac{1}{2},$$

*provided that $(i)$ $\log \det(\mathbf{I} + \mathbf{B}^{\natural\top} \mathbf{B}^\natural / \sigma^2) < \log n$ and $(ii)$ $n \ge n_0$.*

---

[1] Experiments suggest that we may relax this assumption to that $\mathbf{X}_{ij}$ are i.i.d. sub-Gaussian random variables.

One drawback of this bound is the missing role of the sparsity number $k$. This is because Theorem 1 assumes $\mathbf{B}^\natural$ to be perfectly known while sparsity number $k$ only kicks in when $\mathbf{B}^\natural$ needs to be reconstructed. To handle such issue, we take $\text{supp}(\mathbf{B}^\natural)$ into account as well. Then we have

**Theorem 2.** *There exists an integer $n_0 \geq 0$ such that for arbitrary estimators $\widehat{\mathbf{\Pi}}$ and $\widehat{\mathbf{B}}$, we have*

$$\inf_{\widehat{\mathbf{\Pi}},\widehat{\mathbf{B}}} \sup_{\substack{\mathbf{\Pi} \in \mathcal{P}_n, \\ \mathbf{B} \in \mathcal{B}_{n,p,m,k}}} \mathbb{P}_{\mathbf{X},\mathbf{W}}[(\widehat{\mathbf{\Pi}}, \text{supp}(\widehat{\mathbf{B}})) \neq (\mathbf{\Pi}, \text{supp}(\mathbf{B}))] \geq \frac{1}{2},$$

*hold for all $n \geq n_0$, where $\mathcal{P}_n$ denotes the set of all permutation matrices, $\mathcal{B}_{n,p,m,k}$ is defined as the set reading as $\{\mathbf{B} \in \mathbb{R}^{p \times m} : \log \det(\mathbf{I} + \mathbf{B}^\top \mathbf{B}/\sigma^2) \leq \log n + \frac{m \log \binom{p}{k}}{n}\}$, and $\text{supp}(\cdot) \triangleq \{(i,j) : (\cdot)_{i,j} \neq 0\}$ denotes the support set.*

This theorem suggests that for all possible estimators to reconstruct $\mathbf{\Pi}$ and $\text{supp}(\mathbf{B})$, there exists at least one pair $(\mathbf{\Pi}, \mathbf{B})$, $\mathbf{\Pi} \in \mathcal{P}_n, \mathbf{B} \in \mathcal{B}_{n,p,m,k}$ such that the reconstruction error rate will be at least $1/2$. Hence, a reliable correspondence reconstruction requires $\mathbf{B}^\natural$ (a fixed but unknown matrix) to satisfy $\log \det \left(\mathbf{I} + \mathbf{B}^{\natural\top}\mathbf{B}^\natural/\sigma^2\right) > \log n + m \log \binom{p}{k}/n$.

We leave the rigorous proof to the supplementary material and only give an intuitive interpretation, which comes from the coding theory. First, we assume each entry in $\mathbf{B}$ to be binary, i.e., $\mathbf{B}_{ij} \in \{0,1\}$, $(1 \leq i \leq p, 1 \leq j \leq m)$. Thus, the information of $\mathbf{B}$ is fully incorporated in $\text{supp}(\mathbf{B})$. In the following, we will use $\text{supp}(\mathbf{B})$ and $\mathbf{B}$ interchangeably, as they are identical.

Afterwards, we view the sensing relation in (1) as the following transmission process: $(i)$ pair $(\mathbf{\Pi}, \mathbf{B})$ is encoded into the codeword $\mathbf{\Pi}\mathbf{X}\mathbf{B}$; $(ii)$ $\mathbf{\Pi}\mathbf{X}\mathbf{B}$ passes through a Gaussian additive channel; and $(iii)$ one observes $\mathbf{Y}$, from which one would like to obtain $(\mathbf{\Pi}, \mathbf{B})$.

Using the terminology from coding theory, we can compute the corresponding code rate and channel capacity as $\frac{\log n! + m \log \binom{p}{k}}{n}$ and $\log \det(\mathbf{I} + \mathbf{B}^\top \mathbf{B}/\sigma^2)$, respectively. Due to the Shannon theorem, we can expect non-negligible decoding error if the coding rate is greater than the channel capacity, which leads to $\log \det(\mathbf{I} + \mathbf{B}^\top \mathbf{B}/\sigma^2) < \log n + \frac{m \log \binom{p}{k}}{n}$, the formula in Theorem 2.

**Remark 1.** *Note that the minimum SNR requirements can be derived from Theorem 2 as $\log \det(\mathbf{I} + \mathbf{B}^\top \mathbf{B}/\sigma^2)$ is closely related to SNR. When $\mathbf{B}$ is of rank-one, we have $\log \det(\mathbf{I} + \mathbf{B}^\top \mathbf{B}/\sigma^2)$ be $\log(1 + m \cdot \text{SNR})$. When $\mathbf{B}$ is of full-rank and with identical singular values, we have $\log \det(\mathbf{I} + \mathbf{B}^\top \mathbf{B}/\sigma^2)$ be $\text{rank}(\mathbf{B}) \cdot \log(1 + \text{SNR})$.*

Having obtained the information theoretic lower bounds, we will propose a computationally efficient estimator which matches the lower bounds thereof to a good extent.

## 4 ESTIMATOR DESIGN

This section proposes a computationally efficient estimator for the permutation recovery. Denote $\text{thres}(\cdot)$ as the operator which only keeps the element with the largest magnitude in each column, we reconstruct $\mathbf{\Pi}^\natural$ with the *linear assignment problem* (LAP) (Kuhn, 1955; Bertsekas & Castañón, 1992; Burkard et al., 2012) reading as

$$\mathbf{\Pi}^{\text{opt}} = \text{argmax}_{\mathbf{\Pi} \in \mathcal{P}_n} \left\langle \mathbf{\Pi}, \mathbf{Y} \cdot \text{thres}(\mathbf{X}^\top \mathbf{Y})^\top \cdot \mathbf{X}^\top \right\rangle,$$

Once the permutation matrix $\mathbf{\Pi}^{\text{opt}}$ is obtained, we can transform (1) to the previous setting and iteratively recover each $k$-sparse column $\mathbf{B}^\natural_{:,i}$. A formal statement of the algorithm is in Algorithm 1. Here we use Lasso estimator to reconstruct the signal $\mathbf{B}^\natural$, which can be replaced with other estimators, say Dantzig estimator, etc.

**Design intuition.** The design of (2) shares a similar idea of the estimator in Zhang & Li (2020): we would like to approximate the direction of $\mathbf{B}^\natural$ by the product $\mathbf{X}^\top \mathbf{Y}$. However, due to insufficient samples, product $\mathbf{X}^\top \mathbf{Y}$ is poorly aligned with $\mathbf{B}^\natural$, or equivalently, large errors in approximating $\mathbf{B}^\natural$ with $\mathbf{X}^\top \mathbf{Y}$ and weak correlation $\langle \mathbf{B}^\natural, \mathbf{X}^\top \mathbf{Y} \rangle$. To reduce the approximation errors, we apply $\text{thres}(\cdot)$ and set certain entries in $\mathbf{X}^\top \mathbf{Y}$ to zero.

---

**Algorithm 1:** One-step estimator.

---

**Input:** observation $\mathbf{Y}$ and sensing matrix $\mathbf{X}$.
**Output:** pair $(\mathbf{\Pi}^{\mathrm{opt}}, \mathbf{B}^{\mathrm{opt}})$, which is written as

$$\mathbf{\Pi}^{\mathrm{opt}} = \mathrm{argmax}_{\mathbf{\Pi}\in\mathcal{P}_n} \left\langle \mathbf{\Pi}, \mathbf{Y} \cdot \mathsf{thres}(\mathbf{X}^\top\mathbf{Y})^\top \cdot \mathbf{X}^\top \right\rangle, \tag{2}$$

$$\mathbf{B}^{\mathrm{opt}} = \mathrm{argmin}_{\mathbf{B}} (2n)^{-1} \left\| \mathbf{\Pi}^{\mathrm{opt}\top}\mathbf{Y} - \mathbf{X}\mathbf{B} \right\|_{\mathrm{F}}^2 + \lambda_n \|\mathbf{B}\|_1, \tag{3}$$

where $\mathsf{thres}(\cdot)$ applies to each column and thresholds all entries to zero except the one with the largest magnitude, $\mathcal{P}_n$ denotes the set of all possible permutation matrices, $\|\cdot\|_1 \triangleq \sum_{i,j} |(\cdot)_{i,j}|$ denotes the absolute sum of all entries, and $\lambda_n > 0$ is some regularizer coefficient.

---

Note that **we always keep one nonzero entry in the operation $\mathsf{thres}(\cdot)$ regardless of the sparsity number $k$.** This operation is different from almost all the previous works, which ranges from Blumensath & Davies (2009); Foucart (2011) in the literature of compressive sensing to Jain et al. (2013); Yuan et al. (2014); Li et al. (2016) in the literature of optimization, since all these works suggest keeping at least $O(k)$ non-zero elements for a $k$-sparse signal.

More surprisingly, our numerical experiments suggest keeping more non-zero elements are detrimental to the permutation recovery. An illustration is put in Figure 1, from which we observe the SNR required for correct correspondence recovery increases with the number of non-zero elements kept in $\mathbf{X}^\top\mathbf{Y}$.

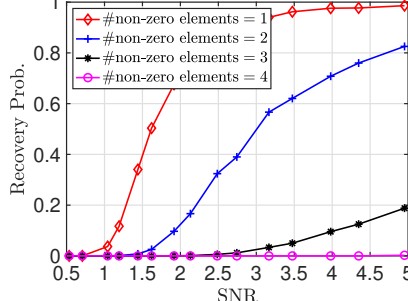

Figure 1: Keeping more non-zero elements in $\mathbf{X}^\top\mathbf{Y}$ deteriorates permutation recovery. $n = 100$, $p = 500$, $h = 25$, $k = 5$, $\mathrm{srank}(\mathbf{B}^\natural) = 100$. **#non-zero elements** refers to the number of non-zero elements kept in each column of $\mathbf{X}^\top\mathbf{Y}$.

**Remark 2.** *We apply the operator $\mathsf{thres}(\cdot)$ to $\mathbf{X}^\top\mathbf{Y}$ to better approximate $\mathbf{B}^\natural$'s direction, or equivalently, increase the correlation $\langle\mathsf{thres}(\mathbf{X}^\top\mathbf{Y}), \mathbf{B}^\natural\rangle$. Due to the insufficient sample number $n$, $\mathbf{X}^\top\mathbf{Y}$ is poorly aligned with $\mathbf{B}^\natural$. Thus, keeping more non-zero elements in $\mathsf{thres}(\cdot)$ leads to a potential decrease of $\langle\mathsf{thres}(\mathbf{X}^\top\mathbf{Y}), \mathbf{B}^\natural\rangle$ and less satisfactory performance.*

### 4.1 MAIN RESULTS

This subsection formally presents our main results.

#### 4.1.1 RESULTS IN RECOVERING $\mathbf{\Pi}^\natural$

First, we study the correspondence recovery. The formal statement is given as

**Theorem 3.** *Suppose that $n \geq n_0$ and $p \geq p_0$, where $n_0, p_0 > 0$ are some positive constants. Provided that $(i)$ $n \gg k(\log n)(\log^2 mnp)$, $(ii)$ $\mathrm{srank}(\mathbf{B}^\natural) \gg k^2 \log^{2(1+\varepsilon_0)} n$, $(iii)$ $h \leq c_0 \cdot n$, and $(iv)$ $\mathsf{SNR} \geq c_1$, we have $\{\mathbf{\Pi}^{\mathrm{opt}} = \mathbf{\Pi}^\natural\}$ with probability at least $1 - c_2 \cdot n^{-c_3}$, where $\varepsilon_0 > 0$ is an arbitrary positive constant and $h \triangleq \mathsf{d}_\mathsf{H}(\mathbf{I}, \mathbf{\Pi}^\natural)$ denotes the number of permuted rows.*

If we assume that for each column its maximum entry's energy is at least a constant proportion of the total energy, i.e., $\inf_j \max_i |\mathbf{B}^\natural_{i,j}|/\|\mathbf{B}^\natural_{:,j}\|_2 \geq \varepsilon_1$ ($1 \leq i \leq p, 1 \leq j \leq m$, we can further relax the requirements on stable rank $\mathrm{srank}(\mathbf{B}^\natural)$. Here $\varepsilon_1 > 0$ is an arbitrarily small positive constant.

**Discussion.** Comparison with previous work is put in Table 1, from which we conclude that **our work gives the first affirmative answer such that $\mathsf{SNR} \geq \Omega(1)$ is sufficient to obtain the correct permutation matrix with insufficient samples**, i.e., $n \ll p$.

In addition, we would like to compare it with the lower bound. To begin with, we discuss the SNR requirement, which is the top priority of our analysis. From Theorem 3, we can see that the correct permutation matrix can be obtained provided that SNR is above some positive constant; meanwhile

Table 1: Comparison with previous works. All results are presented in their best orders, which only hold true in certain regimes. Here $\mathsf{SNR}_{\min}$, $n_{\min}$ and $h_{\max}$ denotes the minimum SNR required for correct permutation recovery, the minimum required sample number and maximum allowed number of permuted rows, respectively. Moreover, the logarithmic term is omitted in $\widetilde{\Omega}(\cdot)$.

| | $\mathsf{SNR}_{\min} (\geq)$ | | $n_{\min}/p \ (\geq)$ | | $h_{\max}/n \ (\leq)$ | |
|---|---|---|---|---|---|---|
| | $m = 1$ | $m \gg 1$ | $m = 1$ | $m \gg 1$ | $m = 1$ | $m \gg 1$ |
| (Pananjady et al., 2018) | $\widetilde{\Omega}(n^c)$ | | $\widetilde{\Omega}(1)$ | | $\widetilde{\Omega}(1)$ | |
| (Slawski & Ben-David, 2019) | $\widetilde{\Omega}(n^c)$ | | $\widetilde{\Omega}(1)$ | | $\widetilde{\Omega}(\log^{-1} n)$ | |
| (Zhang et al., 2022) | | $\widetilde{\Omega}(1)$ | | $\widetilde{\Omega}(1)$ | | $\widetilde{\Omega}\left(\log^{-1} r(\mathbf{B}^\natural)\right)$ |
| (Slawski et al., 2020) | | $\widetilde{\Omega}(1)$ | | $\widetilde{\Omega}(p)$ | | $\widetilde{\Omega}(\log^{-1} n)$ |
| (Zhang & Li, 2020) | $\widetilde{\Omega}(n^c)$ | $\widetilde{\Omega}(1)$ | $\widetilde{\Omega}(1)$ | $\widetilde{\Omega}(\sqrt{p})$ | $\widetilde{\Omega}(1)$ | $\widetilde{\Omega}(1)$ |
| (Zhang & Li, 2021) | $\widetilde{\Omega}(n^c)$ | | $o(1)$ | | $\widetilde{\Omega}(1)$ | |
| **Our Estimator** | | $\widetilde{\Omega}(1)$ | | $o(1)$ | | $\widetilde{\Omega}(1)$ |

Theorem 2 requires SNR $> 0$. This means our SNR requirement has at most a gap of some positive constant with the statistical lower bound.

For the sample number $n$, the lower bound requires $n$ to be at least of order $\Omega(k \log p)$; while Theorem 3 requires $n$ to be $\Omega(k(\log n)(\log^2 mnp))$, which means the lower bound is matched up to some multiplicative logarithmic terms. We conjecture that the required sample number $n$ in Theorem 3 can be further optimized, i.e., the logarithmic terms, with a more delicate analysis.

Moreover, our estimator allows the maximum number of permuted rows to be linearly proportional to the sample number, i.e., $h_{\max} \asymp n$, which is order-optimal.

**Remark 3.** *Compared with Zhang & Li (2020) which only requires* $\mathrm{srank}(\mathbf{B}^\natural)$ *to be above certain positive constant, our estimator requires a larger stable rank* $\mathrm{srank}(\mathbf{B}^\natural)$. *Although we cannot claim that* $\mathrm{srank}(\mathbf{B}^\natural)$ *must be lower bounded by some non-decreasing functions of* $\log n$, *we have a numerical evidence such that* $\mathrm{srank}(\mathbf{B}^\natural)$ *may have to increase with sample number* $n$, *in other words, its lower bound may not be reduced to be some positive constant. Fixing the parameters* $p$, $k$, $h$, *and* $\mathrm{srank}(\mathbf{B}^\natural)$, *we study the impact of sample number* $n$ *on the permutation recovery and put the results in Figure 2. We observe that a larger* $n$ *has a negative impact on the permutation reconstruction once* $n$ *exceeds certain threshold. One possible reason is that the stable rank* $\mathrm{srank}(\mathbf{B}^\natural)$ *is fixed as a constant and violates the requirement* $\mathrm{srank}(\mathbf{B}^\natural) \gg \log^2 n$.

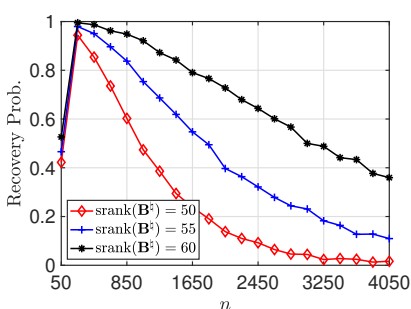

Figure 2: Illustration of the dual role of sample number $n$. We consider the noiseless case (infinite SNR); and set $p = 200$ (signal length), $k = 5$ (sparsity number), and $h = 25$ (number of permuted rows).

### 4.1.2 RESULTS IN RECOVERING $\mathbf{B}^\natural$

Once the ground-truth $\mathbf{\Pi}^\natural$ is obtained, we restore (1) to the traditional model in compressive sensing/sparse recovery (Candès et al., 2006; Candes et al., 2006; Donoho, 2006; Wainwright, 2019). One corollary is given as follows.

**Corollary 1.** *Suppose that* $n \geq n_0$ *and* $p \geq p_0$, *where* $n_0, p_0 > 0$ *are some positive constants. Provided that (i)* $n \gg k(\log n)(\log^2 mnp)$, *(ii)* $\mathrm{srank}(\mathbf{B}^\natural) \gg k^2 \log^{2(1+\varepsilon)} n$, *(iii)* $h \leq c_0 \cdot n$, *and (iv)* $\mathsf{SNR} \geq c_1$. *Setting* $\lambda_n$ *in (3) as* $c_2 \sigma \sqrt{\log p / n}$, *we conclude*

$$\left\| \mathbf{B}^\natural - \mathbf{B}^{\mathrm{opt}} \right\|_{\mathrm{F}} \lesssim \sigma \sqrt{\frac{mk \log p}{n}}$$

*holds with probability exceeding* $1 - c_3 n^{-c_4} - c_5 p^{-c_6}$.

Its proof is a simple combination of Theorem 3 and the previous results in Candès et al. (2006); Candes et al. (2006); Donoho (2006); Wainwright (2019). However, we observe a new phenomenon: the reconstruction error $\left\|\mathbf{B}^{\mathrm{opt}} - \mathbf{B}^{\natural}\right\|_{\mathrm{F}}$ is affected by the signal energy $\left\|\mathbf{B}^{\natural}\right\|_{\mathrm{F}}$ on top of the sensing noise $\sigma^2$. [2] Moreover, we should mention that the above difference will still exist even when we replace (3) with other estimators, say, Dantzig estimator.

## 4.2 PROOF OUTLINE

Due to the space limit, we only give a sketch of our proof ideas and leave the technical details to the supplementary material. Denote $\widehat{\mathbf{B}} = (n-h)^{-1}\mathbf{X}^\top\mathbf{Y}$, our goal is to show

$$\langle \mathbf{Y}, \mathbf{\Pi}^{\natural}\mathbf{X} \cdot \mathsf{thres}(\widehat{\mathbf{B}}) \rangle > \langle \mathbf{Y}, \mathbf{\Pi}\mathbf{X} \cdot \mathsf{thres}(\widehat{\mathbf{B}}) \rangle, \ \ \forall \, \mathbf{\Pi} \neq \mathbf{\Pi}^{\natural} \tag{4}$$

holds with high probability under the settings in Theorem 3.

Same as Zhang & Li (2020), our analysis faces the difficulties brought by $(i)$ combinatorial nature of the problem and $(ii)$ high-order moments of sub-Gaussian random variables. On top of these challenges, we are subject to insufficient samples, i.e., $n \ll p$. These issues are tackled by a combination of relaxation and a tailored leave-one-out analysis, which can be roughly divided into the following three stages.

**Stage I.** We consider the sufficient condition of (4), which reads as

$$\langle \mathbf{Y}_{i,:}, \mathsf{thres}(\widehat{\mathbf{B}})^\top\mathbf{X}_{\pi^{\natural}(i),:} \rangle \geq \langle \mathbf{Y}_{i,:}, \mathsf{thres}(\widehat{\mathbf{B}})^\top\mathbf{X}_{j,:} \rangle, \ \ \forall \, j \neq \pi^{\natural}(i).$$

Re-arranging the terms, we obtain an equivalent form reading as

$$\begin{aligned} \langle \mathbf{B}^{\natural\top}\mathbf{X}_{\pi^{\natural}(i),:}, \mathsf{thres}(\widehat{\mathbf{B}})^\top\mathbf{X}_{\pi^{\natural}(i),:} \rangle \geq \ & \langle \mathbf{B}^{\natural\top}\mathbf{X}_{\pi^{\natural}(i),:}, \mathsf{thres}(\widehat{\mathbf{B}})^\top\mathbf{X}_{j,:} \rangle \\ & + \langle \mathbf{W}_{i,:}, \mathsf{thres}(\widehat{\mathbf{B}})^\top \left( \mathbf{X}_{\pi^{\natural}(i),:} - \mathbf{X}_{j,:} \right) \rangle. \end{aligned} \tag{5}$$

Informally speaking, we first assume that $\mathsf{thres}(\widehat{\mathbf{B}})$ is almost parallel to $\mathbf{B}^{\natural}$; and the dependence of $\mathsf{thres}(\widehat{\mathbf{B}})$ on $\mathbf{X}_{\pi^{\natural}(i),:}, \mathbf{X}_{j,:}$ is negligible. Then, we can approximate (5)'s left-hand side as $\left\|\mathbf{B}^{\natural}\right\|_{\mathrm{F}}^2$ and its right-hand side as a sum of $\mathbf{z}_1\mathbf{B}^{\natural\top}\mathbf{B}^{\natural}\mathbf{z}_2$ and $\sqrt{2}z\left\|\mathbf{B}^{\natural}\right\|_{\mathrm{F}}$ up to some normalization constant, where $\mathbf{z}_1, \mathbf{z}_2 \overset{\mathrm{i.i.d}}{\sim} \mathsf{N}(\mathbf{0}, \mathbf{I}_{p\times p})$ and $\mathbf{z} \overset{\mathrm{i.i.d}}{\sim} \mathsf{N}(\mathbf{0}, \mathbf{I}_{m\times m})$ are Gaussian random vectors. Easily, we can see that (5) holds with high probability provided the SNR is sufficiently large. In the following two stages, we will verify the above two assumptions, that is, $(i)$ $\angle(\mathsf{thres}(\widehat{\mathbf{B}}), \mathbf{B}^{\natural})$ is small and $(ii)$ the dependence between $\mathsf{thres}(\widehat{\mathbf{B}})$ and $\mathbf{X}_{\pi^{\natural}(i),:}, \mathbf{X}_{j,:}$ is negligible.

**Stage II.** We would like to lower-bound the inner product $\langle \mathbf{B}^{\natural}, \mathsf{thres}(\widehat{\mathbf{B}}) \rangle$. Denote $\widehat{\boldsymbol{\beta}}$ as the corresponding column in $\widehat{\mathbf{B}}$, we can express $\langle \mathbf{B}^{\natural}, \mathsf{thres}(\widehat{\mathbf{B}}) \rangle$ as $\sum_{\boldsymbol{\beta}^{\natural}\in\{\mathbf{B}^{\natural}_{:,\ell}\}_{1\leq\ell\leq m}} \langle \boldsymbol{\beta}^{\natural}, \mathsf{thres}(\widehat{\boldsymbol{\beta}}) \rangle$. From the definition of $\mathsf{thres}(\cdot)$, we notice that $\mathsf{thres}(\widehat{\boldsymbol{\beta}})$ only has one non-zero entry. W.l.o.g. we assume its index is one and hence have

$$\langle \boldsymbol{\beta}^{\natural}, \mathsf{thres}(\widehat{\boldsymbol{\beta}}) \rangle = \boldsymbol{\beta}_1^{\natural}\widehat{\boldsymbol{\beta}}_1 = (\boldsymbol{\beta}_1^{\natural})^2 + \boldsymbol{\beta}_1^{\natural}(\widehat{\boldsymbol{\beta}}_1 - \boldsymbol{\beta}_1^{\natural}) \geq (\boldsymbol{\beta}_1^{\natural})^2 - \max_i |\boldsymbol{\beta}_i^{\natural}| \cdot \|\boldsymbol{\beta}^{\natural} - \widehat{\boldsymbol{\beta}}\|_{\infty}.$$

We $(i)$ upper-bound $\max_i |\boldsymbol{\beta}_i^{\natural}|$ and $\|\boldsymbol{\beta}^{\natural} - \widehat{\boldsymbol{\beta}}\|_{\infty}$; and $(ii)$ lower-bound $|\boldsymbol{\beta}_1^{\natural}|$. Part $(i)$ is quite standard and part $(ii)$ lies in analyzing the event $|\widehat{\boldsymbol{\beta}}_1| \geq \max_j |\widehat{\boldsymbol{\beta}}_j|$, which is due to the definition of $\mathsf{thres}(\cdot)$.

**Stage III.** We would like to show the dependence between $\mathsf{thres}(\widehat{\mathbf{B}})$ and rows $\mathbf{X}_{\pi^{\natural}(i),:}$ and $\mathbf{X}_{j,:}$ is negligible. This is accomplished by a tailored leave-one-out analysis. For each row indices pair $(\pi^{\natural}(i), j)$, we construct a perturbed matrix $\widehat{\mathbf{B}}^{(\pi^{\natural}(i),j)}$ by replacing the rows $\mathbf{X}_{\pi^{\natural}(i),:}, \mathbf{X}_{j,:}$ with their i.i.d. substitutes. Easily, we can verify that $\widehat{\mathbf{B}}^{(\pi^{\natural}(i),j)}$ is independent from the rows $\mathbf{X}_{\pi^{\natural}(i),:}, \mathbf{X}_{j,:}$ as the latter are not involved in $\widehat{\mathbf{B}}^{(\pi^{\natural}(i),j)}$. Meanwhile, we have $\widehat{\mathbf{B}}^{(\pi^{\natural}(i),j)}$ exhibit a similar behavior as $\widehat{\mathbf{B}}$ as they share almost identical components. Actually, this is the basic idea of leave-one-out technique (Chen et al., 2020; Sur et al., 2019; El Karoui, 2013; 2018; Cai et al., 2021). Compared with these works, our construction method has the following characteristics

---

[2]This corollary has one requirement on SNR, which is affected by both $\left\|\mathbf{B}^{\natural}\right\|_{\mathrm{F}}$ and $\sigma^2$. In addition, we may have $\left\|\mathbf{B}^{\mathrm{opt}} - \mathbf{B}^{\natural}\right\|_{\mathrm{F}}$ be directly linked to $\left\|\mathbf{B}^{\natural}\right\|_{\mathrm{F}}$, as a wrong correspondence can be obtained with low SNR.

- The number of replaced rows in our method varies for different pair of row indices. Meanwhile, the replacement number is fixed in the above mentioned works. For a better explanation, we refer the readers to our constructed *leave-one-out* samples $\widetilde{\mathbf{B}}_{(\cdot)}$ in the appendix.

- We modify the operator $\mathsf{thres}(\cdot)$ in approximating $\mathsf{thres}(\widehat{\mathbf{B}})$. While the previous works usually keep the operator $\mathsf{thres}(\cdot)$ intact, we approximate it with the operator $(\cdot)_{\mathsf{imax}}$, which denotes the positions of non-zero elements in $\mathsf{thres}(\widehat{\mathbf{B}})$. In other words, the positions of the non-zero elements we keep in the *leave-one-out* samples $\widetilde{\mathbf{B}}^{(\cdot)}$ are determined by $\mathsf{thres}(\widehat{\mathbf{B}})$ rather than $\mathsf{thres}(\widetilde{\mathbf{B}}^{(\cdot)})$. In our analysis, we can see this step is essential in controlling the approximation errors. Otherwise, the approximation error can be considerably large, since $\mathsf{thres}(\widehat{\mathbf{B}})$ may not share the same support set with $\mathsf{thres}(\widetilde{\mathbf{B}}^{(\cdot)})$, let alone their $\ell_2$ differences.

The explanation thereof is a simplified version of our proof technique. The technical details, which are put in the supplementary material, can be different from the above however follow the same spirit.

Moreover, we want to discuss our algorithm's computational complexity: in the first step for permutation recovery, our estimator only requires one matrix multiplication and thresholding operation on top of the operations in the oracle estimator; in the second step for sparse signal recovery, our estimator needs one additional matrix multiplication when compared with the work without permutation.

## 5 NUMERICAL RESULTS

This section presents the numerical experiments to verify our main theorem, to put more specifically, Theorem 3: we would like to prove the correct permutation can be obtained, i.e., $\{\mathbf{\Pi}^{\mathsf{opt}} = \mathbf{\Pi}^{\natural}\}$, with $n \ll p$ and $\mathsf{SNR} \geq c$. We only present the numerical results on the synthetic data here and defer those on the real-world data to the supplementary material.

**Experiment setting with Gaussian distribution.** We let $\mathbf{X}_{ij} \overset{\text{i.i.d}}{\sim} \mathsf{N}(0,1)$ and pick the sample number $n$ to be $\{100, 150\}$ and set $h = {}^n\!/_4$. We vary the signal length $p$ to be $\{500, 600\}$. Then we set the sparsity number $k$ within the region $\{10, 15, 20\}$. And the stable rank $\mathsf{srank}(\mathbf{B}^{\natural})$ is within the range $\{150, 200, 250\}$. The corresponding simulation results can be found in Figure 3.

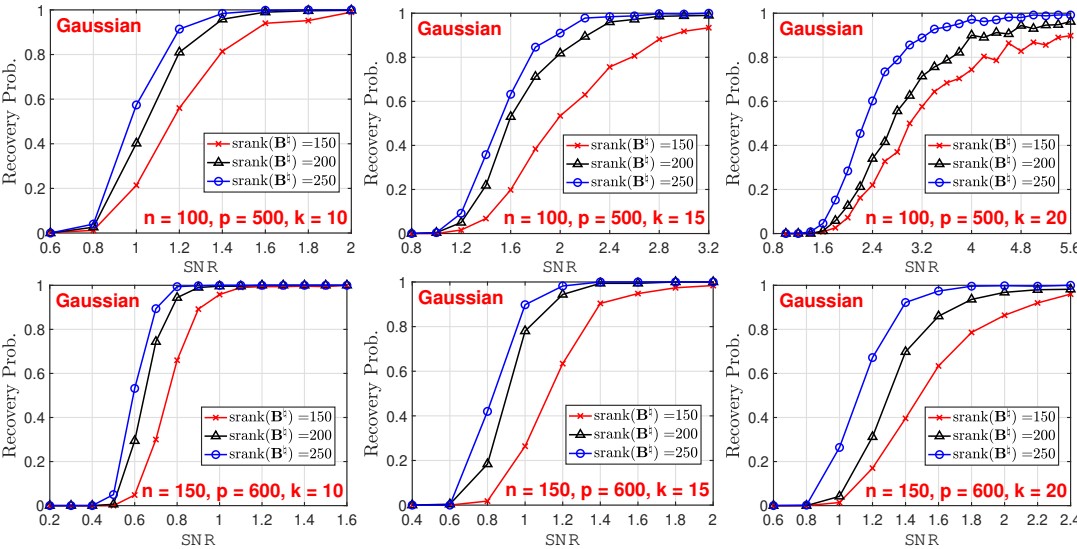

Figure 3: Simulated recovery rate $\mathbb{P}(\widehat{\mathbf{\Pi}} = \mathbf{\Pi}^{\natural})$ with $n = \{100, 150\}$, $p \in \{500, 600\}$, $h \in \{25, 37\}$, and $\mathbf{X}_{ij} \overset{\text{i.i.d}}{\sim} \mathsf{N}(0, 1)$, with respect to SNR.

**Discussion w.r.t** ${}^n\!/_p$**.** First, we confirm our theory such that correct permutation can be obtained with insufficient samples, i.e., $n \ll p$. In addition, we notice that the permutation recovery becomes easy with a larger ${}^n\!/_p$: the first row in Figure 3 is with ${}^n\!/_p = 0.2$ while the second row is with ${}^n\!/_p = 0.25$.

We can verify the SNR for the correct permutation recovery is smaller in the second row than that for the first row. However, we should stress that this conclusion may not hold provided that $\text{srank}(\mathbf{B}^\natural)$ is not sufficiently large. More details are referred to Figure 2.

**Discussion w.r.t. sparsity number $k$.** We vary the sparsity number $k$ to be within $\{10, 15, 20\}$. We conclude a large sparsity number $k$ can make the permutation recovery more difficult. For example, consider the case when $(n, p, \text{srank}(\mathbf{B}^\natural)) = (100, 500, 200)$. When $k = 10$, correct permutation requires $\text{SNR} \geq 1.4$; when $k = 15$, correct permutation needs $\text{SNR} \geq 2.2$; and when $k = 20$, correct permutation requires $\text{SNR} \geq 4$. The same conclusion holds for other cases as well. [3]

**Experiment setting with sub-Gaussian distribution.** In addition to the Gaussian setting, we also evaluate our estimator's performance when $\mathbf{X}_{ij}$ being sub-Gaussian. Here, we pick $\mathbf{X}_{ij}$ to be i.i.d. Rademacher random variables such that $\mathbb{P}(\mathbf{X}_{ij} = \pm 1) = 1/2$. The corresponding results are put in Figure 4, from which we can observe a similar pattern as that of the Gaussian setting.

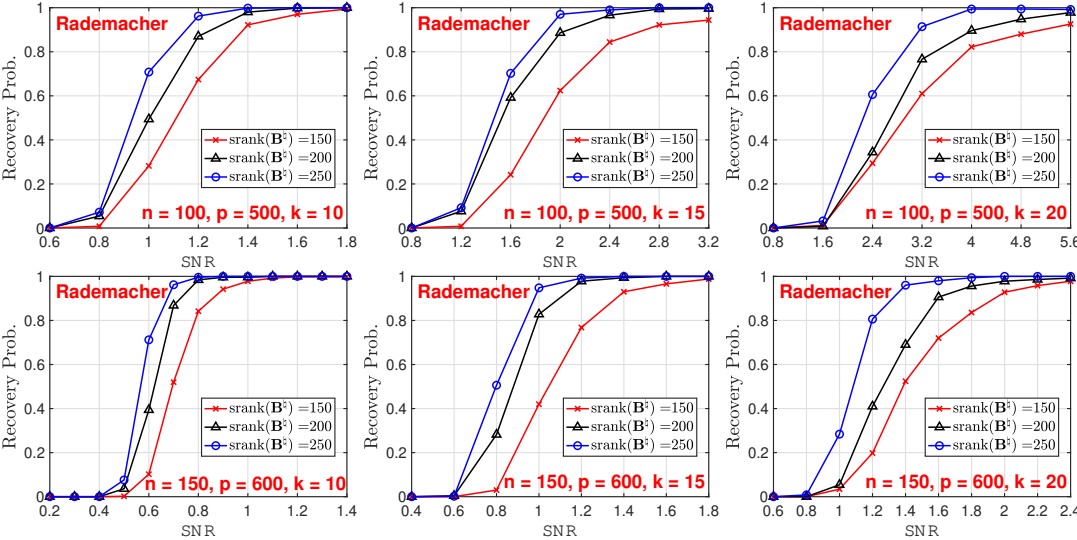

Figure 4: Simulated recovery rate $\mathbb{P}(\widehat{\mathbf{\Pi}} = \mathbf{\Pi}^\natural)$ with $n = \{100, 150\}$, $p \in \{500, 600\}$, $h \in \{25, 37\}$, and $\mathbf{X}_{ij} \overset{\text{i.i.d}}{\sim}$ Rademacher, i.e., $\mathbb{P}(\mathbf{X}_{ij} = -1) = \mathbb{P}(\mathbf{X}_{ij} = 1) = 1/2$, w.r.t. SNR.

## 6 CONCLUSION

In this paper, we studied the unlabeled sparse recovery with multiple measurements (i.e., $m > 1$) for the first time. To begin with, we investigated the lower bounds on the sample number $n$ and the SNR. Furthermore, we proposed a simple yet effective estimator, which restores the permutation matrix via a linear assignment problem. We proved that our estimator can obtain the correct correspondence information when SNR is above certain positive constant and required sample number $n$ is in linear dependence with sparsity number $k$. In addition, we discovered multiple phenomena that are seldom encountered before: $(i)$ keeping more non-zero elements in $\text{thres}(\cdot)$ deteriorates permutation recovery; and $(ii)$ increasing sample number $n$ plays a dual role in reconstructing the permutation. In the course of analyzing our estimator's performance and explaining the above phenomena, we developed a tailored version of the leave-one-out technique, which involves an adaptive number of replaced elements and simultaneous modification of the threshold operator. Moreover, we provided numerical experiments to corroborate our claims and showed our estimator can reliably reconstruct the permutation matrix even when the entries $\mathbf{X}_{ij}$ are sub-Gaussian random variables.

---

[3]This result is consistent with Theorem 2 as $\mathcal{B}_{n,p,m,k}$ covers a broader class of matrices $\mathbf{B}$ when $k$ increases from 15 to 20, which implies a larger SNR is required for correct permutation recovery.

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

# A PROOF OF THEOREM 2

*Proof.* The proof technique is a combination of that in Zhang et al. (2022) and Zhang & Li (2021). First we put uniform distribution as the prior of $\mathbf{\Pi}$, i.e., $\mathbb{P}(\mathbf{\Pi}^{\natural} = \mathbf{\Pi}_{\mathrm{samp}}) = |\mathcal{P}_n|^{-1}$, where $\mathbf{\Pi}_{\mathrm{samp}}$ is an arbitrary fixed permutation matrix and $\mathcal{P}_n$ denotes the set of all possible permutation matrices. In addition, we introduce distributions for the support set of $\mathbf{B}$. For an arbitrary column $\mathbf{B}_{:,\ell}$, we assume its support set to be uniformly distributed among $\binom{p}{k}$ possible patterns. Easily, we can verify the relation

$$\sup_{\mathbf{\Pi},\mathbf{B}} \mathbb{P}_{\mathbf{X},\mathbf{W}}\left[(\mathbf{\Pi}, \mathrm{supp}(\mathbf{B})) \neq (\widehat{\mathbf{\Pi}}, \mathrm{supp}(\widehat{\mathbf{B}}))\right] \geq \mathbb{P}_{\mathbf{X},\mathbf{W},\mathbf{\Pi},\mathbf{B}}\left[(\mathbf{\Pi}, \mathrm{supp}(\mathbf{B})) \neq (\widehat{\mathbf{\Pi}}, \mathrm{supp}(\widehat{\mathbf{B}}))\right]. \quad (6)$$

Since $\inf_{\widehat{\mathbf{\Pi}},\widehat{\mathbf{B}}}$ can be safely added to the left-hand side of (6), our goal becomes lower-bounding $\mathbb{P}_{\mathbf{X},\mathbf{W},\mathbf{\Pi},\mathbf{B}}\left[(\mathbf{\Pi}, \mathrm{supp}(\mathbf{B})) \neq (\widehat{\mathbf{\Pi}}, \mathrm{supp}(\widehat{\mathbf{B}}))\right]$. Adopting the proof technique used in Theorem 2.10.1 in Cover & Thomas (2012), we consider the entropy $\mathsf{H}(\mathbf{\Pi}, \mathrm{supp}(\mathbf{B}))$, which can be computed as

$$\mathsf{H}(\mathbf{\Pi}, \mathrm{supp}(\mathbf{B})) \overset{\textcircled{1}}{=} \mathsf{H}(\mathbf{\Pi}) + \mathsf{H}(\mathrm{supp}(\mathbf{B})) \overset{\textcircled{2}}{=} \log n! + m \cdot \log \binom{p}{k}, \quad (7)$$

where in $\textcircled{1}$ we use the independent among $\mathbf{\Pi}$ and $\mathrm{supp}(\mathbf{B})$, and in $\textcircled{2}$ we use the fact $|\mathcal{P}_n| = n!$ and $|\mathrm{supp}(\mathbf{B})| = \binom{p}{k}^m$. Meanwhile, we have the relation

$$\begin{aligned}
\mathsf{H}(\mathbf{\Pi}, \mathrm{supp}(\mathbf{B})) &\overset{\textcircled{3}}{=} \mathsf{H}(\mathbf{\Pi}, \mathrm{supp}(\mathbf{B}) \mid \mathbf{X}) \\
&\overset{\textcircled{4}}{=} \underbrace{\mathsf{H}(\mathbf{\Pi}, \mathrm{supp}(\mathbf{B}) \mid \mathbf{X}, \widehat{\mathbf{\Pi}}, \mathrm{supp}(\widehat{\mathbf{B}}))}_{\triangleq \zeta_1} + \underbrace{\mathsf{I}(\mathbf{\Pi}, \mathrm{supp}(\mathbf{B}); \widehat{\mathbf{\Pi}}, \mathrm{supp}(\widehat{\mathbf{B}}) \mid \mathbf{X})}_{\triangleq \zeta_2}. \quad (8)
\end{aligned}$$

where $\textcircled{3}$ is due to the independence between $\mathbf{X}$ and $\mathbf{\Pi}, \mathbf{B}$; and $\textcircled{4}$ is because of the definition of the conditional entropy and mutual information. The proof is thus complete by separately bounding $\eta_1$ and $\eta_2$.

**Analysis of $\zeta_1$.** We upper-bound $\zeta_1$ with Fano's inequality (Cover & Thomas, 2012, Theorem 2.10.1), which proceeds as

$$\begin{aligned}
\zeta_1 &\leq \mathsf{H}(\mathbf{\Pi}, \mathrm{supp}(\mathbf{B}) \mid \widehat{\mathbf{\Pi}}, \mathrm{supp}(\widehat{\mathbf{B}})) \\
&\leq 1 + \log\left(|(\mathbf{\Pi}, \mathrm{supp}(\mathbf{B}))|\right) \cdot \mathbb{P}_{\mathbf{X},\mathbf{W},\mathbf{\Pi},\mathbf{B}}\left[(\mathbf{\Pi}, \mathrm{supp}(\mathbf{B})) \neq (\widehat{\mathbf{\Pi}}, \mathrm{supp}(\widehat{\mathbf{B}}))\right]. \quad (9)
\end{aligned}$$

**Analysis of $\zeta_2$.** Due to the Markov property of $(\mathbf{\Pi}, \mathrm{supp}(\mathbf{B})) \to \mathbf{Y} \to (\widehat{\mathbf{\Pi}}, \mathrm{supp}(\widehat{\mathbf{B}}))$, we invoke the data-processing inequality (Cover & Thomas, 2012, Thm. 2.8.1) and conclude

$$\zeta_2 \leq \mathsf{I}(\mathbf{\Pi}, \mathrm{supp}(\mathbf{B}); \mathbf{Y} \mid \mathbf{X}).$$

Invoking the definition of conditional mutual information, we have

$$\begin{aligned}
\mathsf{I}(\mathbf{\Pi}, \mathrm{supp}(\mathbf{B}); \mathbf{Y} \mid \mathbf{X}) &= \mathbb{E}_{\mathbf{X},\mathbf{W},\mathbf{\Pi}}\left[\mathsf{h}(\mathbf{Y} \mid \mathbf{X} = \boldsymbol{x})) - \mathsf{h}(\mathbf{Y} \mid \mathbf{\Pi}, \mathrm{supp}(\mathbf{B}), \mathbf{X} = \boldsymbol{x})\right] \\
&\overset{\textcircled{5}}{\leq} \frac{1}{2} \log \det\left(\mathbb{E}_{\mathbf{X},\mathbf{W},\mathbf{\Pi}} \mathbf{Y}\mathbf{Y}^\top\right) - \frac{mn}{2} \log \sigma^2,
\end{aligned}$$

where in $\textcircled{5}$ we use the property (Cover & Thomas, 2012, Theorem 8.6.5)

$$\mathsf{h}(\mathbf{Z}) \leq \frac{1}{2} \log \det \mathrm{Cov}(\mathbf{Z}) \leq \frac{1}{2} \log \det \mathbb{E}\left(\mathbf{Z}\mathbf{Z}^\top\right),$$

for a random vector $\mathbf{Z}$ with finite covariance matrix $\mathrm{Cov}(\mathbf{Z})$, and the entropy for a Gaussian random vector. Following the same procedure as in (Zhang et al., 2022, Lemma 11), we have

$$\log \det \mathbb{E}_{\mathbf{X},\mathbf{W},\mathbf{\Pi}} \mathbf{Y}\mathbf{Y}^\top = nm \cdot \log \sigma^2 + n \cdot \log \det\left(\mathbf{I} + \mathbf{B}^\top\mathbf{B}/\sigma^2\right),$$

which further yields to

$$\mathsf{I}(\mathbf{\Pi}, \mathrm{supp}(\mathbf{B}); \mathbf{Y} \mid \mathbf{X}) \leq \frac{n}{2} \log \det\left(\mathbf{I} + \mathbf{B}^\top\mathbf{B}/\sigma^2\right).$$

**Summary.** Combing (7), (8), and (9) then leads to a lower-bound on $\mathbb{P}_{\mathbf{X},\mathbf{W},\mathbf{\Pi},\mathbf{B}}\big[(\mathbf{\Pi},\operatorname{supp}(\mathbf{B})) \neq (\widehat{\mathbf{\Pi}},\operatorname{supp}(\widehat{\mathbf{B}}))\big]$ reading as

$$\mathbb{P}_{\mathbf{X},\mathbf{W},\mathbf{\Pi},\mathbf{B}}\big[(\mathbf{\Pi},\operatorname{supp}(\mathbf{B})) \neq (\widehat{\mathbf{\Pi}},\operatorname{supp}(\widehat{\mathbf{B}}))\big] \geq \frac{\log n! + m\dot{\log}\binom{p}{k} - 1 - (n/2)\log\det\big(\mathbf{I} + \mathbf{B}^\top\mathbf{B}/\sigma^2\big)}{\log\left(|(\mathbf{\Pi},\operatorname{supp}(\mathbf{B}))|\right)}.$$

Easily, we can verify $\mathbb{P}_{\mathbf{X},\mathbf{W},\mathbf{\Pi},\mathbf{B}}\big[(\mathbf{\Pi},\operatorname{supp}(\mathbf{B})) \neq (\widehat{\mathbf{\Pi}},\operatorname{supp}(\widehat{\mathbf{B}}))\big]$ is lower bounded by $1/2$ given the assumptions in Theorem 2 and thus complete the proof.

$\square$

## B  PROOF OF THEOREM 3

We define $\widehat{\mathbf{B}}$ as $(n-h)^{-1}\mathbf{X}^\top\mathbf{Y}$ and define operator $\mathsf{imax}(i)$ as $\operatorname{argmax}_j|\widehat{\mathbf{B}}_{j,i}|$ $(1 \leq i \leq m)$ for each column in $\widehat{\mathbf{B}}$, which returns the index of the entry with the largest magnitude. With a slight abuse of notation, we denote $\widehat{\mathbf{B}}_{\mathsf{imax}} = \mathsf{thres}(\widehat{\mathbf{B}})$. The benefits of this notation will be seen shortly.

In addition, we define the error $\mathcal{E}_{\mathrm{err}}$ as

$$\mathcal{E}_{\mathrm{err}} = \left\{\exists\, \mathbf{\Pi},\ \text{s.t.}\ \ \left\langle \mathbf{Y}, \mathbf{\Pi}\mathbf{X}\cdot\widehat{\mathbf{B}}_{\mathsf{imax}}\right\rangle \geq \left\langle\mathbf{Y}, \mathbf{\Pi}^\natural\mathbf{X}\cdot\widehat{\mathbf{B}}_{\mathsf{imax}}\right\rangle\right\}. \tag{10}$$

According to the sensing relation such that $\mathbf{Y} = \mathbf{\Pi}^\natural\mathbf{X}\mathbf{B}^\natural + \mathbf{W}$, we rewrite (10) as

$$\left\langle\mathbf{\Pi}^\natural\mathbf{X}\mathbf{B}^\natural + \mathbf{W}, \mathbf{\Pi}\mathbf{X}\cdot\widehat{\mathbf{B}}_{\mathsf{imax}}\right\rangle \leq \left\langle\mathbf{\Pi}^\natural\mathbf{X}\mathbf{B}^\natural + \mathbf{W}, \mathbf{\Pi}^\natural\mathbf{X}\cdot\widehat{\mathbf{B}}_{\mathsf{imax}}\right\rangle,$$

and would like to show it holds with probability near zero. The major technical difficulties come from the fact that $\widehat{\mathbf{B}}$ is correlated with sensing matrix $\mathbf{X}$, which introduces high-order moments. The solution of such challenge is broadly divided into the following two parts.

**Part I: Relaxation of error event.** We first relax the error event $\mathcal{E}_{\mathrm{err}}$

$$\mathcal{E}_{\mathrm{err}} \subseteq \underbrace{\left\{\left\langle\mathbf{Y}_{i,:}, \widehat{\mathbf{B}}_{\mathsf{imax}}^\top\mathbf{X}_{\pi^\natural(i),:}\right\rangle \leq \left\langle\mathbf{Y}_{i,:}, \widehat{\mathbf{B}}_{\mathsf{imax}}^\top\mathbf{X}_{j,:}\right\rangle,\ \exists\ j \neq \pi^\natural(i)\right\}}_{\triangleq\,\mathcal{E}_{\mathrm{err\text{-}relax}}}, \tag{11}$$

which means $\mathbb{P}(\mathcal{E}_{\mathrm{err}}) \leq \mathbb{P}(\mathcal{E}_{\mathrm{err\text{-}relax}})$.

**Part II: Decoupling dependence via a modified leave-one-out technique.** To decompose the dependence between $\widehat{\mathbf{B}}$ and the rows $\mathbf{X}_{\pi^\natural(i),:}$ and $\mathbf{X}_{j,:}$, we modify the leave-one-out technique and construct a perturbed matrix $\widehat{\mathbf{B}}^{(\pi^\natural(i),j)}$, which shares almost identical statistical behaviors as $\widehat{\mathbf{B}}$. Before delving into the technical details, we first provide a glimpse of the construction idea. Recalling the definition of $\widehat{\mathbf{B}}$, which is written as

$$\widehat{\mathbf{B}} = \frac{1}{n-h}\left(\sum_{\ell=1}^n\mathbf{X}_{\ell,:}\mathbf{X}_{\pi^\natural(\ell),:}^\top\right)\mathbf{B}^\natural + \frac{\mathbf{X}^\top\mathbf{W}}{n-h},$$

we construct the perturbed matrix $\widehat{\mathbf{B}}^{(\pi^\natural(i),j)}$ by replacing the corresponding rows with their i.i.d. samples. The detailed construction method is stated as follows.

To begin with, we draw i.i.d. samples for each rows of $\mathbf{X}_{i,:}$ and denote it as $\widetilde{\mathbf{X}}_{i,:}$ $(1 \leq i \leq n)$. Similarly we draw samples $\widetilde{\mathbf{W}}_{j,:}$ for each row in $\mathbf{W}$, $1 \leq j \leq n$. For arbitrary indices $\pi^\natural(i)$ and $j$ such that $j \neq \pi^\natural(i)$, we create the samples $\widehat{\mathbf{B}}^{(\pi^\natural(i),j)}$ as (assume $i \neq \pi^\natural(i)$ and $j \neq \pi^\natural(j)$)

$$\begin{aligned}
\widehat{\mathbf{B}}^{(\pi^\natural(i),j)} =\ & (n-h)^{-1}\left(\sum_{\ell,\pi^\natural(\ell)\neq\pi^\natural(i),j}\mathbf{X}_{\ell,:}\mathbf{X}_{\pi^\natural(\ell),:}^\top\right)\mathbf{B}^\natural \\
& + (n-h)^{-1}\left(\widetilde{\mathbf{X}}_{i,:}\widetilde{\mathbf{X}}_{\pi^\natural(i),:}^\top + \widetilde{\mathbf{X}}_{\pi^\natural(i),:}\widetilde{\mathbf{X}}_{\pi^\natural(\pi^\natural(i)),:}^\top + \widetilde{\mathbf{X}}_{j,:}\widetilde{\mathbf{X}}_{\pi^\natural(j),:}^\top + \widetilde{\mathbf{X}}_{\pi^{\natural-1}(j),:}\widetilde{\mathbf{X}}_{j,:}^\top\right)\mathbf{B}^\natural \\
& + (n-h)^{-1}\left(\sum_{\ell\neq\pi^\natural(i),i}\mathbf{X}_{\ell,:}\mathbf{W}_{\ell,:}^\top + \widetilde{\mathbf{X}}_{\pi^\natural(i),:}\widetilde{\mathbf{W}}_{\pi^\natural(i),:}^\top + \widetilde{\mathbf{X}}_{i,:}\widetilde{\mathbf{W}}_{i,:}^\top\right). 
\end{aligned} \tag{12}$$

Provided that $i = \pi^\natural(i)$, we can simplify the summaries $\widetilde{\mathbf{X}}_{i,:}\widetilde{\mathbf{X}}^\top_{\pi^\natural(i),:} + \widetilde{\mathbf{X}}_{\pi^\natural(i),:}\widetilde{\mathbf{X}}^\top_{\pi^\natural(\pi^\natural(i)),:}$ and $\widetilde{\mathbf{X}}_{\pi^\natural(i),:}\widetilde{\mathbf{W}}^\top_{\pi^\natural(i),:} + \widetilde{\mathbf{X}}_{i,:}\widetilde{\mathbf{W}}^\top_{i,:}$ in the above construction as the terms $\widetilde{\mathbf{X}}_{i,:}\widetilde{\mathbf{X}}^\top_{i,:}$ and $\widetilde{\mathbf{X}}_{i,:}\widetilde{\mathbf{W}}^\top_{i,:}$, respectively. Similarly, we will simplify $\widetilde{\mathbf{X}}_{j,:}\widetilde{\mathbf{X}}^\top_{\pi^\natural(j),:} + \widetilde{\mathbf{X}}_{\pi^{\natural-1}(j),:}\widetilde{\mathbf{X}}^\top_{j,:}$ as $\mathbf{X}_{j,:}\mathbf{X}^\top_{j,:}$ when $j = \pi^\natural(j)$.

With the above construction method, easily we can verify that $\widehat{\mathbf{B}}^{(\pi^\natural,j)}$ is independent of the rows $\mathbf{X}_{\pi^\natural(i),:}, \mathbf{X}_{j,:}$ and $\mathbf{W}_{i,:}$ as they are not involved in $\widehat{\mathbf{B}}^{(\pi^\natural,j)}$. Before delving into the technical details, we first collect all required notations.

### B.1 NOTATIONS

Define the following events

$$\mathcal{E}_1 \triangleq \left\{ \left| \frac{\sum_{\ell=\pi^\natural(\ell)} \mathbf{X}^2_{\ell,i}}{n-h} - 1 \right| \le c_0 \sqrt{\frac{\log(np)}{n-h}}, \ \forall\, 1 \le i \le p \right\};$$

$$\mathcal{E}_2(\boldsymbol{\beta}) \triangleq \left\{ \frac{\sum_{\ell=\pi^\natural(\ell)} \mathbf{X}_{\ell,i} \left\langle \mathbf{X}_{\ell,\backslash i}\boldsymbol{\beta}_{\backslash i} \right\rangle}{n-h} \lesssim \sqrt{\frac{\log(mnp)}{n-h}} \|\boldsymbol{\beta}_{\backslash i}\|_2, \ \forall\, 1 \le i \le p \right\};$$

$$\mathcal{E}_3(\boldsymbol{\beta}) \triangleq \left\{ \frac{\sum_{\ell\neq\pi^\natural(\ell)} \mathbf{X}_{\ell,i} \left\langle \mathbf{X}_{\pi^\natural(\ell),:},\boldsymbol{\beta} \right\rangle}{n-h} \lesssim \frac{\sqrt{h}\log(mnp)}{n-h} \|\boldsymbol{\beta}\|_2, \ \forall\, 1 \le i \le p \right\};$$

$$\mathcal{E}_4 \triangleq \left\{ \frac{\sum_{\ell=1}^n \mathbf{X}^\top_{\ell,i}\mathbf{W}_{\ell,j}}{n-h} \lesssim \frac{\sigma\sqrt{n}\log(mnp)}{n-h}, \ \forall\, 1 \le i \le p, 1 \le j \le m \right\};$$

$$\mathcal{E}_5 \triangleq \left\{ \left\| (\widehat{\mathbf{B}}_{\mathsf{imax}} - \widehat{\mathbf{B}}^{(\pi^\natural(i),j)}_{\mathsf{imax}})^\top \boldsymbol{x} \right\|_2 \lesssim \frac{\log^{3/2}(np)}{n-h}\left\|\!\left\|\mathbf{B}^\natural\right\|\!\right\|_{\mathrm{F}} + \frac{\sigma(\log np)\sqrt{m(\log mn)}}{n-h}, \ \forall\, 1 \le \pi^\natural(i) \neq j \le p \right\},$$

where $\boldsymbol{\beta} \in \mathbb{R}^p$ is an arbitrary column of $\mathbf{B}^\natural_{\ell,:}$ $(1 \le \ell \le m)$, $\boldsymbol{\beta}_{\backslash i} \in \mathbb{R}^p$ denotes its copy with the $i$th entry being set to be zero, and $\boldsymbol{x} \in \mathbb{R}^p$ denotes an arbitrary row in matrix $\mathbf{X}$, which follows Gaussian distribution $\mathsf{N}(\mathbf{0}, \mathbf{I}_{p\times p})$. Note that $\boldsymbol{x}$ is not necessarily independent from $\widehat{\mathbf{B}}$ and $\widehat{\mathbf{B}}^{(\pi^\natural(i),j)}$.

For an arbitrary event $E$, we denote its complement as $\overline{E}$. In addition, we define matrix $\mathbf{M}^{(\pi^\natural(i),j)}$ as $\mathbf{B}^{\natural\top}\widehat{\mathbf{B}}^{(\pi^\natural(i),j)}_{\mathsf{imax}}$. For the notational simplicity, we drop the superscript $\pi^\natural(i)$ and $j$ in $\mathbf{M}^{(\pi^\natural(i),j)}$ when there is no ambiguity. The following context provides the technical details and a diagram representing the dependence among all lemmas is put in Figure 5.

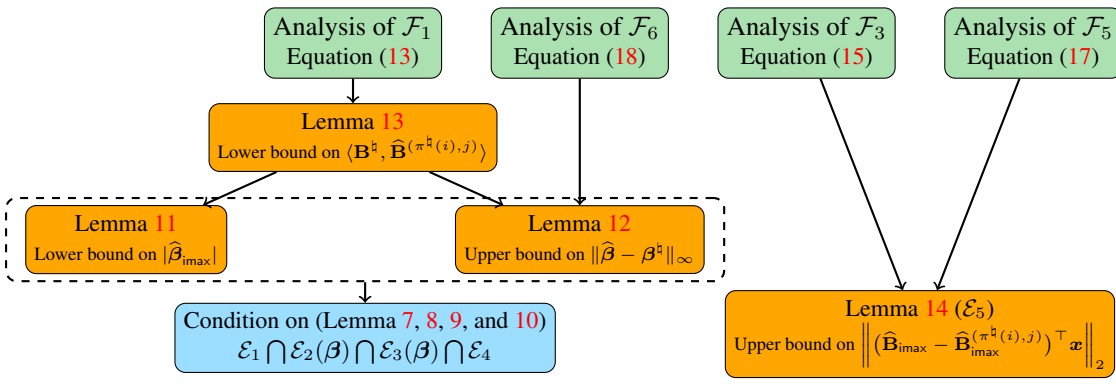

Figure 5: Dependence diagram of lemmas.

### B.2 Proof of Theorem 3

*Proof.* The proof can be broadly divided into three stages.

**Stage I.** To begin with, we prove the relation $\mathcal{E}_{\text{err}} \subseteq \mathcal{E}_{\text{err-relax}}$, whose definition can be found in (11). Conditional on $\overline{\mathcal{E}}_{\text{err-relax}}$, we have

$$\left\langle \mathbf{Y}_{i,:}, \widehat{\mathbf{B}}_{\text{imax}}^{\top}\mathbf{X}_{\pi^{\natural}(i),:} \right\rangle > \left\langle \mathbf{Y}_{i,:}, \widehat{\mathbf{B}}_{\text{imax}}^{\top}\mathbf{X}_{j,:} \right\rangle, \ \forall \ j \neq \pi^{\natural}(i).$$

Thus we conclude

$$\left\langle \mathbf{Y}, \mathbf{\Pi}^{\natural}\mathbf{X} \cdot \widehat{\mathbf{B}}_{\text{imax}} \right\rangle > \left\langle \mathbf{Y}, \mathbf{\Pi}\mathbf{X} \cdot \widehat{\mathbf{B}}_{\text{imax}} \right\rangle, \ \forall \ \mathbf{\Pi} \neq \mathbf{\Pi}^{\natural},$$

which suggests $\overline{\mathcal{E}}_{\text{err-relax}}$ will automatically lead to $\overline{\mathcal{E}}_{\text{err}}$, in other words, $\overline{\mathcal{E}}_{\text{err-relax}} \subseteq \overline{\mathcal{E}}_{\text{err}}$. Hence, we could upper bound $\mathbb{P}(\mathcal{E}_{\text{err}})$ by $\mathbb{P}(\mathcal{E}_{\text{err-relax}})$.

**Stage II.** Regarding the relation $\langle \mathbf{Y}_{i,:}, \widehat{\mathbf{B}}_{\text{imax}}^{\top}\mathbf{X}_{\pi^{\natural}(i),:} \rangle \leq \langle \mathbf{Y}_{i,:}, \widehat{\mathbf{B}}_{\text{imax}}^{\top}\mathbf{X}_{j,:} \rangle$, we can recast it as

$$\underbrace{\langle \mathbf{B}^{\natural\top}\mathbf{X}_{\pi^{\natural}(i),:}, \widehat{\mathbf{B}}_{\text{imax}}^{(\pi^{\natural}(i),j)\top}\mathbf{X}_{\pi^{\natural}(i),:} \rangle}_{\eta_1^{(\pi^{\natural}(i),j)}} \leq \underbrace{\langle \mathbf{B}^{\natural\top}\mathbf{X}_{\pi^{\natural}(i),:}, \widehat{\mathbf{B}}_{\text{imax}}^{(\pi^{\natural}(i),j)\top}\mathbf{X}_{j,:} \rangle}_{\eta_2^{(\pi^{\natural}(i),j)}}$$

$$+ \underbrace{\langle \mathbf{B}^{\natural\top}\mathbf{X}_{\pi^{\natural}(i),:}, \left(\widehat{\mathbf{B}}_{\text{imax}} - \widehat{\mathbf{B}}_{\text{imax}}^{(\pi^{\natural}(i),j)}\right)^{\top}\left(\mathbf{X}_{j,:} - \mathbf{X}_{\pi^{\natural}(i),:}\right) \rangle}_{\eta_3^{(\pi^{\natural}(i),j)}}$$

$$+ \underbrace{\left\langle \mathbf{W}_{i,:}, \widehat{\mathbf{B}}_{\text{imax}}^{(\pi^{\natural}(i),j)\top}\left(\mathbf{X}_{j,:} - \mathbf{X}_{\pi^{\natural}(i),:}\right) \right\rangle}_{\eta_4^{(\pi^{\natural}(i),j)}}$$

$$+ \underbrace{\left\langle \mathbf{W}_{i,:}, \left(\widehat{\mathbf{B}}_{\text{imax}} - \widehat{\mathbf{B}}_{\text{imax}}^{(\pi^{\natural}(i),j)}\right)^{\top}\left(\mathbf{X}_{j,:} - \mathbf{X}_{\pi^{\natural}(i),:}\right) \right\rangle}_{\eta_5^{(\pi^{\natural}(i),j)}}.$$

We should emphasize that the subscript imax are solely determined by $\widehat{\mathbf{B}}$ rather than its perturbed partner $\widehat{\mathbf{B}}^{(\pi^{\natural}(i),j)}$. This is to ensure $\widehat{\mathbf{B}}$ and $\widehat{\mathbf{B}}^{(\pi^{\natural}(i),j)}$ share the same support set. In the following analysis, we will see this property plays an important role in bounding the difference between $\widehat{\mathbf{B}}_{\text{imax}}$ and $\widehat{\mathbf{B}}_{\text{imax}}^{(\pi^{\natural}(i),j)}$, which is contained in $\eta_3^{(\pi^{\natural}(i),j)}$ and $\eta_5^{(\pi^{\natural}(i),j)}$. The following context separately studies each term $\eta_{\ell}^{(\pi^{\natural}(i),j)}$ ($1 \leq \ell \leq 5$). First we define quantities $\Delta_{\ell}^{(\pi^{\natural}(i),j)}$ ($1 \leq \ell \leq 5$) as

$$\Delta_1^{(\pi^{\natural}(i),j)} \triangleq \frac{\left\|\!\left\|\mathbf{B}^{\natural}\right\|\!\right\|_{\text{F}}^2}{k} - \frac{m\sigma^2(\log mnp)^2}{n} - \frac{\sqrt{m}\sigma\log(mnp)}{\sqrt{n}}\left\|\!\left\|\mathbf{B}^{\natural}\right\|\!\right\|_{\text{F}} - \frac{c_0\log n}{\sqrt{\text{srank}(\mathbf{B}^{\natural})}}\left\|\!\left\|\mathbf{B}^{\natural}\right\|\!\right\|_{\text{F}}\left\|\!\left\|\widehat{\mathbf{B}}_{\text{imax}}^{(\pi^{\natural}(i),j)}\right\|\!\right\|_{\text{F}};$$

$$\Delta_2^{(\pi^{\natural}(i),j)} \triangleq \log n \left\|\!\left\|\mathbf{B}^{\natural}\right\|\!\right\|_{\text{F}}\left\|\!\left\|\widehat{\mathbf{B}}_{\text{imax}}^{(\pi^{\natural}(i),j)}\right\|\!\right\|_{\text{F}} / \sqrt{\text{srank}(\mathbf{B}^{\natural})};$$

$$\Delta_3^{(\pi^{\natural}(i),j)} \triangleq \frac{\left\|\!\left\|\mathbf{B}^{\natural}\right\|\!\right\|_{\text{F}}^2\sqrt{\log n}(\log np)^{3/2}}{n} + \frac{\left\|\!\left\|\mathbf{B}^{\natural}\right\|\!\right\|_{\text{F}}\sigma\sqrt{m\log n}(\log mn)^{1/2}(\log np)}{n};$$

$$\Delta_4^{(\pi^{\natural}(i),j)} \triangleq \sigma(\log n)\left\|\!\left\|\widehat{\mathbf{B}}_{\text{imax}}^{(\pi^{\natural}(i),j)}\right\|\!\right\|_{\text{F}};$$

$$\Delta_5^{(\pi^{\natural}(i),j)} \triangleq \frac{\sqrt{m\log n}\left(\log^{3/2}np\right)}{n-h}\left\|\!\left\|\mathbf{B}^{\natural}\right\|\!\right\|_{\text{F}}\sigma + \frac{m\sigma^2(\log np)\sqrt{(\log n)(\log mn)}}{n-h}.$$

In addition, we define the events $\mathcal{F}_\ell$ $(1 \leq \ell \leq 6)$ as

$$\mathcal{F}_1 \triangleq \left\{ \eta_1^{(\pi^\natural(i),j)} \gtrsim \Delta_1^{(\pi^\natural(i),j)}, \ \forall \ 1 \leq \pi^\natural(i) \neq j \leq p \right\}; \tag{13}$$

$$\mathcal{F}_2 \triangleq \left\{ |\eta_2^{(\pi^\natural(i),j)}| \lesssim \Delta_2^{(\pi^\natural(i),j)}, \ \forall \ 1 \leq \pi^\natural(i) \neq j \leq p \right\}; \tag{14}$$

$$\mathcal{F}_3 \triangleq \left\{ |\eta_3^{(\pi^\natural(i),j)}| \lesssim \Delta_3^{(\pi^\natural(i),j)}, \ \forall \ 1 \leq \pi^\natural(i) \neq j \leq p \right\}; \tag{15}$$

$$\mathcal{F}_4 \triangleq \left\{ |\eta_4^{(\pi^\natural(i),j)}| \lesssim \Delta_4^{(\pi^\natural(i),j)}, \ \forall \ 1 \leq \pi^\natural(i) \neq j \leq p \right\}; \tag{16}$$

$$\mathcal{F}_5 \triangleq \left\{ |\eta_5^{(\pi^\natural(i),j)}| \lesssim \Delta_5^{(\pi^\natural(i),j)}, \ \forall \ 1 \leq \pi^\natural(i) \neq j \leq p \right\}, \tag{17}$$

$$\mathcal{F}_6 \triangleq \left\{ \left\|\left\| \widehat{\mathbf{B}}_{\mathsf{imax}}^{(\pi^\natural(i),j)} - \mathbf{B}_{\mathsf{imax}}^\natural \right\|\right\|_{\mathrm{F}} \lesssim \frac{\log(mnp)}{\sqrt{n}} \left\|\left\| \mathbf{B}^\natural \right\|\right\|_{\mathrm{F}} + \frac{\sigma \log(mnp)\sqrt{m}}{\sqrt{n}}, \ \forall \ 1 \leq \pi^\natural(i) \neq j \leq p \right\}. \tag{18}$$

In Lemma 1, Lemma 2, Lemma 3, Lemma 4, Lemma 5, and Lemma 12, we will show all the above events, namely, $\mathcal{F}_\ell$ $(1 \leq \ell \leq 6)$, hold with probability approaching one.

**Stage III.** Given the assumptions in Theorem 3, we will verify the relation $\bigcap_{\ell=1}^6 \mathcal{F}_\ell \subseteq \mathcal{E}_{\text{err-relax}}$. Considering the difference $\Delta_1^{(\pi^\natural(i),j)} - \sum_{\ell=2}^5 \Delta_\ell^{(\pi^\natural(i),j)}$, we can lower bound it as

$$\frac{\Delta_1^{(\pi^\natural(i),j)}}{m\sigma^2} - \sum_{\ell=2}^5 \frac{\Delta_\ell^{(\pi^\natural(i),j)}}{m\sigma^2} \geq \mathsf{SNR} \underbrace{\left( \frac{1}{k} - \frac{\sqrt{\log n}\log^{3/2}(np)}{n} - \frac{2\log n}{\sqrt{\mathrm{srank}(\mathbf{B}^\natural)}} \right)}_{\zeta_1}$$

$$- \sqrt{\mathsf{SNR}} \underbrace{\left( \frac{\sqrt{\log n}(\log np)\left( \sqrt{\log mn} + \sqrt{\log np} \right)}{n} + \frac{\log mnp}{\sqrt{n}} + \frac{\log n}{\sqrt{m}} + \frac{2(\log n)(\log mnp)}{\sqrt{n} \cdot \mathrm{srank}(\mathbf{B}^\natural)} \right)}_{\zeta_{1/2}}$$

$$- \underbrace{\left( \frac{\log n(\log mnp)}{\sqrt{mn}} + \frac{\log^2(mnp) + \sqrt{\log n}(\log mn)(\log np)}{n} \right)}_{\zeta_0}.$$

Under the assumptions in Theorem 3, we have

$$\zeta_1 \asymp k^{-1};$$

$$\zeta_{1/2} \asymp \frac{\log mnp}{\sqrt{n}} + \frac{\log n}{\sqrt{m}};$$

$$\zeta_0 \asymp \frac{\log n(\log mnp)}{\sqrt{mn}} + \frac{\log^2(mnp)}{n},$$

which leads to

$$\frac{\Delta_1^{(\pi^\natural(i),j)}}{m\sigma^2} - \sum_{\ell=2}^5 \frac{\Delta_\ell^{(\pi^\natural(i),j)}}{m\sigma^2} \geq \zeta_1 \cdot \mathsf{SNR} - \zeta_{1/2}\sqrt{\mathsf{SNR}} - \zeta_0$$

$$\overset{\text{①}}{\gtrsim} \frac{1}{k} - \frac{1}{k\sqrt{\log n}} - \frac{1}{k\log^\varepsilon n} - \frac{1}{k^2 \log^{\varepsilon+1/2} n} - \frac{1}{k^2 \log n} > 0,$$

where in ① we use the relation $m \geq \mathrm{rank}(\mathbf{B}^\natural) \geq \mathrm{srank}(\mathbf{B}^\natural) \gg k^2 \log^{2(1+\varepsilon)} n$. Then we conclude

$$\sum_{\ell=2}^5 |\eta_\ell^{(\pi^\natural(i),j)}| \leq \sum_{\ell=2}^5 \Delta_\ell^{(\pi^\natural(i),j)} \leq \Delta_1^{(\pi^\natural(i),j)} \leq \eta_1^{(\pi^\natural(i),j)}, \ \ \forall \ 1 \leq \pi^\natural(i) \neq j \leq p,$$

which means we will automatically obtain $\mathcal{E}_{\text{err-relax}}$ when assuming $\bigcap_{\ell=1}^6 \mathcal{F}_\ell$, in other words, $\bigcap_{\ell=1}^6 \mathcal{F}_\ell \subseteq \mathcal{E}_{\text{err-relax}}$. Hence, we can complete the proof by $\mathbb{P}\left( \mathcal{E}_{\text{err}} \right) \leq \mathbb{P}\left( \mathcal{E}_{\text{err-relax}} \right) \leq \sum_{\ell=1}^6 \mathbb{P}(\overline{\mathcal{F}}_\ell)$.

$\square$

**Lemma 1.** *Conditional on the intersection of events $\mathcal{E}_1 \bigcap_{\ell=1}^m \mathcal{E}_2(\mathbf{B}_{\ell,:}^\natural) \bigcap_{\ell=1}^m \mathcal{E}_3(\mathbf{B}_{\ell,:}^\natural) \bigcap \mathcal{E}_4$, we have $\mathbb{P}(\mathcal{F}_1) \geq 1 - c_0 n^{-c_1}$ provided $n \gg k \log^2(mnp)$.*

*Proof.* Recalling the definition of $\mathbf{M}$, i.e., $\mathbf{M} \triangleq \mathbf{B}^{\natural\top} \widehat{\mathbf{B}}_{\mathsf{imax}}^{(\pi^\natural(i),j)}$, we divide the proof procedure as

- **Step I.** Condition on $\mathbf{M}$, we have

$$\eta_1^{(\pi^\natural(i),j)} \geq \mathrm{Tr}(\mathbf{M}) - \frac{c_0 \log n}{\sqrt{\mathrm{srank}(\mathbf{B}^\natural)}} \left\|\!\left\|\mathbf{B}^\natural\right\|\!\right\|_{\mathrm{F}} \left\|\!\left\|\widehat{\mathbf{B}}_{\mathsf{imax}}^{(\pi^\natural(i),j)}\right\|\!\right\|_{\mathrm{F}}, \ \ \forall 1 \leq \pi^\natural(i) \neq j \leq n,$$

  hold with probability exceeding $1 - n^{-c}$.

- **Step II.** Provided $n \gg k \log^2(mnp)$, we have

$$\mathrm{Tr}(\mathbf{M}) \gtrsim \frac{1}{k} \left\|\!\left\|\mathbf{B}^\natural\right\|\!\right\|_{\mathrm{F}}^2 - \frac{m\sigma^2 (\log mnp)^2}{n} - \frac{\sqrt{m}\sigma \log(mnp)}{\sqrt{n}} \left\|\!\left\|\mathbf{B}^\natural\right\|\!\right\|_{\mathrm{F}},$$

  condition on $\mathcal{E}_1 \bigcap_{\ell=1}^m \mathcal{E}_2(\mathbf{B}_{\ell,:}^\natural) \bigcap_{\ell=1}^m \mathcal{E}_3(\mathbf{B}_{\ell,:}^\natural) \bigcap \mathcal{E}_4$.

For the clarify of presentation, we defer the proof of **Step II** to Lemma 13 and focus on **Step I**. Due to the construction of $\widehat{\mathbf{B}}^{(\pi^\natural,j)}$ in (12), we conclude $\widehat{\mathbf{B}}^{(\pi^\natural,j)}$ is independent with row $\mathbf{X}_{\pi^\natural(i),:}$. Hence we can first condition on $\mathbf{M}$ and rewrite term $\eta_1$ in terms of a quadratic product $\boldsymbol{x}^\top \mathbf{M}\boldsymbol{x}$, where $\boldsymbol{x} \in \mathbb{R}^p$ is a random vector satisfying $\boldsymbol{x} \in \mathsf{N}(\mathbf{0}, \mathbf{I}_{p \times p})$.

With Hanson-Wright inequality (c.f. Theorem 6.2.1 in Vershynin (2018)), we have

$$\mathbb{P}\left(|\eta_1^{(\pi^\natural(i),j)} - \mathbb{E}\eta_1^{(\pi^\natural(i),j)}| \geq t, \ \exists\, 1 \leq \pi^\natural(i) \neq j \leq n\right)$$

$$\leq n^2 \cdot \mathbb{P}\left(\left|\boldsymbol{x}^\top \mathbf{M}\boldsymbol{x} - \mathrm{Tr}(\mathbf{M})\right| \geq t\right) \leq n^2 \cdot 2\exp\left(-c_0 \left(\frac{t}{\|\mathbf{M}\|_{\mathrm{OP}}} \wedge \frac{t^2}{\|\mathbf{M}\|_{\mathrm{F}}^2}\right)\right).$$

Setting $t = \log n\|\mathbf{M}\|_{\mathrm{F}}$, we have

$$\eta_1^{(\pi^\natural(i),j)} \geq \mathbb{E}\eta_1^{(\pi^\natural(i),j)} - c_0 \log n\|\mathbf{M}\|_{\mathrm{F}} = \mathrm{Tr}(\mathbf{M}) - c_0 \log n\|\mathbf{M}\|_{\mathrm{F}}, \ \forall 1 \leq \pi^\natural(i) \neq j \leq n,$$

holds with probability $1 - 2n^{-c}$. Then we complete the proof by showing

$$\|\mathbf{M}\|_{\mathrm{F}} \leq \left\|\!\left\|\mathbf{B}^\natural\right\|\!\right\|_{\mathrm{OP}} \left\|\!\left\|\widehat{\mathbf{B}}_{\mathsf{imax}}^{(\pi^\natural(i),j)}\right\|\!\right\|_{\mathrm{F}} \overset{①}{=} \frac{1}{\sqrt{\mathrm{srank}(\mathbf{B}^\natural)}} \left\|\!\left\|\mathbf{B}^\natural\right\|\!\right\|_{\mathrm{F}} \left\|\!\left\|\widehat{\mathbf{B}}_{\mathsf{imax}}^{(\pi^\natural(i),j)}\right\|\!\right\|_{\mathrm{F}},$$

where ① is due to the definition of stable rank.

$\square$

**Lemma 2.** *We have $\mathbb{P}(\mathcal{F}_2) \geq 1 - 4n^{-c}$ where event $\mathcal{F}_2$ is defined in (14).*

*Proof.* First we fix the indices $\pi^\natural(i)$ and $j$ such that $\pi^\natural(i) \neq j$. Due to the independence across the rows of $\mathbf{X}$, we conclude

$$\mathbb{P}\left(|\boldsymbol{x}^\top \mathbf{M}\boldsymbol{y}| \gtrsim \log n\|\mathbf{M}\|_{\mathrm{F}}\right) \leq \mathbb{P}\left(|\boldsymbol{x}^\top \mathbf{M}\boldsymbol{y}| \gtrsim \log n\|\mathbf{M}\|_{\mathrm{F}}, \ \|\mathbf{M}\boldsymbol{y}\|_2 \lesssim \sqrt{\log n}\|\mathbf{M}\|_{\mathrm{F}}\right)$$

$$+ \mathbb{P}\left(\|\mathbf{M}\boldsymbol{y}\|_2 \gtrsim \sqrt{\log n}\|\mathbf{M}\|_{\mathrm{F}}\right)$$

$$\overset{①}{\leq} \mathbb{P}\left(|\boldsymbol{x}^\top \mathbf{M}\boldsymbol{y}| \gtrsim \sqrt{\log n}\|\mathbf{M}\boldsymbol{y}\|_2\right) + 2n^{-c} \leq 4n^{-c},$$

where $\mathbf{M}$ is defined as $\mathbf{B}^{\natural\top}\widehat{\mathbf{B}}_{\mathsf{imax}}^{(\pi^\natural(i),j)}$ and in ① we have the rotation invariance of Gaussian random vector. Regarding the event $\mathcal{F}_2$, we invoke the union bound and complete the proof as $\mathbb{P}(\mathcal{F}_2) \leq n^2 \cdot 4n^{-c} = 4n^{-c'}$.

$\square$

**Lemma 3.** *Condition on $\mathcal{E}_5$, we conclude $\mathbb{P}\left(\mathcal{F}_3|\mathcal{E}_5\right) \geq 1 - 2n^{-c}$, where $\mathcal{F}_3$ is defined in* (15).

*Proof.* First we recall the definition of $\eta_3$, which is written as

$$\eta_3 \triangleq \left\langle \mathbf{B}^{\natural\top}\mathbf{X}_{i,:}, \left(\widehat{\mathbf{B}}_{\mathsf{imax}} - \widehat{\mathbf{B}}_{\mathsf{imax}}^{(\pi^\natural(i),j)}\right)^\top (\mathbf{X}_{j,:} - \mathbf{X}_{i,:})\right\rangle.$$

We begin the proof as

$$\mathbb{P}\left(\overline{\mathcal{F}}_3|\mathcal{E}_5\right) \leq \mathbb{P}\left(\|\mathbf{B}^{\natural\top}\mathbf{X}_{i,:}\|_2 \gtrsim \log n\|\|\mathbf{B}^\natural\|\|_{\mathrm{F}}^2, \, \exists\, 1 \leq i \leq n|\mathcal{E}_5\right)$$
$$+ \mathbb{P}\left(\overline{\mathcal{F}}_3, \, \|\mathbf{B}^{\natural\top}\mathbf{X}_{i,:}\|_2 \lesssim \log n\|\|\mathbf{B}^\natural\|\|_{\mathrm{F}}^2, \, \forall\, 1 \leq i \leq n|\mathcal{E}_5\right).$$

For the first term, we invoke the Hanson-Wright inequality (c.f. Theorem 6.2.1 in Vershynin (2018)), which leads to

$$\mathbb{P}\left(\|\mathbf{B}^{\natural\top}\mathbf{X}_{i,:}\|_2 \gtrsim \log n\|\|\mathbf{B}^\natural\|\|_{\mathrm{F}}^2, \, \exists\, 1 \leq i \leq n|\mathcal{E}_5\right)$$
$$\leq n \cdot \mathbb{P}\left(\left|\|\mathbf{B}^{\natural\top}\mathbf{X}_{i,:}\|_2^2 - \mathbb{E}\|\mathbf{B}^{\natural\top}\mathbf{X}_{i,:}\|_2^2\right| \gtrsim \log n\|\|\mathbf{B}^\natural\|\|_{\mathrm{F}}^2|\mathcal{E}_5\right)$$
$$\leq 2n \cdot \exp\left(-c\left(\frac{\log^2 n\|\|\mathbf{B}^\natural\|\|_{\mathrm{F}}^4}{\|\mathbf{B}^{\natural\top}\mathbf{B}^\natural\|_{\mathrm{F}}^2} \wedge \frac{\log n\|\|\mathbf{B}^\natural\|\|_{\mathrm{F}}^2}{\|\mathbf{B}^{\natural\top}\mathbf{B}^\natural\|_{\mathrm{OP}}}\right)\right) \leq 2n^{-c}.$$

For the second term, we will prove it to be zero. This is because

$$|\eta_3^{(\pi^\natural(i),j)}| \leq \|\mathbf{B}^{\natural\top}\mathbf{X}_{\pi^\natural(i),:}\|_2 \cdot \left\|\left(\widehat{\mathbf{B}}_{\mathsf{imax}} - \widehat{\mathbf{B}}_{\mathsf{imax}}^{(i,j)}\right)^\top (\mathbf{X}_{j,:} - \mathbf{X}_{\pi^\natural(i),:})\right\|_2$$
$$\overset{①}{\lesssim} \sqrt{\log n}\|\|\mathbf{B}^\natural\|\|_{\mathrm{F}} \cdot \frac{\log^{3/2}(np)}{n-h}\|\|\mathbf{B}^\natural\|\|_{\mathrm{F}} + \frac{\sigma(\log np)\sqrt{m(\log mn)}}{n-h},$$

for all $1 \leq \pi^\natural(i) \neq j \leq n$, where in ① we condition on $\mathcal{E}_5$ and $\|\mathbf{B}^{\natural\top}\mathbf{X}_{\pi^\natural(i),:}\|_2 \lesssim \sqrt{\log n}\|\|\mathbf{B}^\natural\|\|_{\mathrm{F}}$. $\square$

**Lemma 4.** *We have $\mathbb{P}(\mathcal{F}_4) \geq 1 - 4n^{-c}$.*

*Proof.* First we fix the indices $\pi^\natural(i)$ and $j$ such that $\pi^\natural(i) \neq j$. Invoke the definition of $\widehat{\mathbf{B}}_{\mathsf{imax}}^{(\pi^\natural(i),j)}$, we conclude $\mathbf{W}_{i,:}$, $\widehat{\mathbf{B}}_{\mathsf{imax}}^{(\pi^\natural(i),j)}$, $\mathbf{X}_{j,:}$, and $\mathbf{X}_{\pi^\natural(i),:}$ are independent with each other. Hence we can complete the proof as

$$\mathbb{P}\left(|\eta_4^{(\pi^\natural(i),j)}| \gtrsim \sigma(\log n)\|\widehat{\mathbf{B}}_{\mathsf{imax}}^{(\pi^\natural(i),j)}\|_{\mathrm{F}}\right)$$
$$\leq \mathbb{P}\left(\|\widehat{\mathbf{B}}_{\mathsf{imax}}^{(\pi^\natural(i),j)\top}(\mathbf{X}_{j,:} - \mathbf{X}_{\pi^\natural(i),:})\|_2 \gtrsim \sqrt{\log n}\|\widehat{\mathbf{B}}_{\mathsf{imax}}^{(\pi^\natural(i),j)}\|_{\mathrm{F}}\right)$$
$$+ \mathbb{P}\left(|\eta_4| \gtrsim \sigma(\log n)\|\widehat{\mathbf{B}}_{\mathsf{imax}}^{(\pi^\natural(i),j)}\|_{\mathrm{F}}, \|\widehat{\mathbf{B}}_{\mathsf{imax}}^{(\pi^\natural(i),j)\top}(\mathbf{X}_{j,:} - \mathbf{X}_{\pi^\natural(i),:})\|_2 \lesssim \sqrt{\log n}\|\widehat{\mathbf{B}}_{\mathsf{imax}}^{(\pi^\natural(i),j)}\|_{\mathrm{F}}\right)$$
$$\leq 2n^{-c} + \mathbb{P}\left(|\eta_4| \gtrsim \sigma\sqrt{\log n}\|\widehat{\mathbf{B}}_{\mathsf{imax}}^{(\pi^\natural(i),j)\top}(\mathbf{X}_{j,:} - \mathbf{X}_{\pi^\natural(i),:})\|_2\right) \overset{①}{\leq} 4n^{-c},$$

where in ① we use the tail bound for Gaussian random variable. The proof is then completed with the union bound such that

$$\mathbb{P}\left(\overline{\mathcal{F}}_4\right) \leq n^2\mathbb{P}\left(|\eta_4^{(\pi^\natural(i),j)}| \gtrsim \sigma(\log n)\|\widehat{\mathbf{B}}_{\mathsf{imax}}^{(\pi^\natural(i),j)}\|_{\mathrm{F}}\right) \leq 4n^2 \cdot n^{-c} = 1 - 4n^{-c'}.$$

$\square$

**Lemma 5.** *Condition on $\mathcal{E}_5$, we conclude $\mathbb{P}(\mathcal{F}_5|\mathcal{E}_5) \geq 1 - 2n^{-c}$.*

*Proof.* The proof is in a similar form of that for Lemma 3. First we decompose the probability $\mathbb{P}(\overline{\mathcal{F}}_5|\mathcal{E}_5)$ as

$$
\mathbb{P}(\overline{\mathcal{F}}_5|\mathcal{E}_5) \leq \mathbb{P}(\|\mathbf{W}_{i,:}\|_2 \gtrsim \sigma\sqrt{m\log n}, \ \exists\, 1 \leq i \leq n|\mathcal{E}_5) + \mathbb{P}\left(\overline{\mathcal{F}}_5, \ \|\mathbf{W}_{i,:}\|_2 \lesssim \sigma\sqrt{m\log n}, \ \forall\, 1 \leq i \leq n|\mathcal{E}_5\right)
$$

$$
\leq 2n \cdot n^{-c} + \mathbb{P}\left(\overline{\mathcal{F}}_5, \ \|\mathbf{W}_{i,:}\|_2 \lesssim \sigma\sqrt{m\log n}, \ \forall\, 1 \leq i \leq n|\mathcal{E}_5\right),
$$

where the last inequality is due to the tail bound for the Gaussian random variable. We complete the proof by showing the second probability is zero. This is because

$$
|\eta_5^{(\pi^\natural(i),j)}| \leq \|\mathbf{W}_{i,:}\|_2 \cdot \left\|\left(\widehat{\mathbf{B}}_{\mathrm{imax}} - \widehat{\mathbf{B}}_{\mathrm{imax}}^{(i,j)}\right)^\top (\mathbf{X}_{j,:} - \mathbf{X}_{i,:})\right\|_2
$$

$$
\lesssim \frac{\sqrt{m\log n}\left(\log^{3/2} np\right)}{n-h}\|\!|\mathbf{B}^\natural|\!\|_{\mathrm{F}}\sigma + \frac{m\sigma^2(\log np)\sqrt{(\log n)(\log mn)}}{n-h}.
$$

$\square$

**Lemma 6.** *Conditional on the intersection of events $\mathcal{E}_1 \bigcap_{\ell=1}^m \mathcal{E}_2(\mathbf{B}_{\ell,:}^\natural) \bigcap_{\ell=1}^m \mathcal{E}_3(\mathbf{B}_{\ell,:}^\natural) \bigcap \mathcal{E}_4$, we have $\mathbb{P}(\mathcal{F}_5) = 1$.*

*Proof.* We complete the proof by

$$
\left\|\widehat{\mathbf{B}}_{\mathrm{imax}}^{(\pi^\natural(i),j)} - \mathbf{B}_{\mathrm{imax}}^\natural\right\|_{\mathrm{F}}^2 = \sum_{\boldsymbol{\beta}^\natural}\left\|\widehat{\boldsymbol{\beta}}_{\mathrm{imax}}^{(\pi^\natural(i),j)} - \boldsymbol{\beta}_{\mathrm{imax}}^\natural\right\|_2^2 \overset{\text{①}}{=} \sum_{\boldsymbol{\beta}^\natural}\left\|\widehat{\boldsymbol{\beta}}_{\mathrm{imax}}^{(\pi^\natural(i),j)} - \boldsymbol{\beta}_{\mathrm{imax}}^\natural\right\|_\infty^2
$$

$$
\overset{\text{②}}{\lesssim} \frac{\log^2(mnp)}{n}\|\!|\mathbf{B}^\natural|\!\|_{\mathrm{F}}^2 + \frac{m\sigma^2\log^2(mnp)}{n},
$$

where ① is due to the fact such that $\widehat{\boldsymbol{\beta}}_{\mathrm{imax}}^{(\pi^\natural(i),j)} - \boldsymbol{\beta}^\natural$ has only one non-zero element, and ② is due to Lemma 12.

$\square$

## B.3 SUPPORTING LEMMAS

**Lemma 7.** *We have $\mathbb{P}(\mathcal{E}_1) \geq 1 - 2n^{-c_1}p^{-c_2}$ when $n - h \gg \log np$, and $n, p$ are sufficiently large.*

*Proof.* Due to the Hanson-Wright inequality (c.f. Theorem 6.2.1 in Vershynin (2018)), we have

$$
\mathbb{P}(\mathcal{E}_1) \leq 2p\exp\left[-c\left(\frac{(n-h)\log(np)}{\|\!|\mathbf{I}_{(n-h)\times(n-h)}|\!\|_{\mathrm{F}}^2} \wedge \frac{\sqrt{(n-h)\log(np)}}{\|\!|\mathbf{I}_{(n-h)\times(n-h)}|\!\|_{\mathrm{OP}}}\right)\right] \overset{\text{①}}{\leq} 2p \cdot n^{-c}p^{-c} = 2n^{-c_1}p^{-c_2},
$$

where in ① we use the fact $n - h \gg \log(np)$. $\square$

**Lemma 8.** *For a fixed $\boldsymbol{\beta} \in \mathbb{R}^p$, we have $\mathbb{P}(\mathcal{E}_2(\boldsymbol{\beta})) \geq 1 - pe^{-c_0(n-h)} - n^{-c_1}m^{-c_2}p^{-c_3}$.*

*Proof.* To begin with, we construct the sensing matrix $\mathbf{X}_{\mathcal{S}}$ by concatenating all rows $\mathbf{X}_{\ell,:}$ such that $\ell = \pi^\natural(\ell)$. With union bound, we can upper bound $\mathbb{P}\left(\overline{\mathcal{E}}_2(\boldsymbol{\beta})\right)$ as

$$
\mathbb{P}\left(\overline{\mathcal{E}}_2(\boldsymbol{\beta})\right) \leq \underbrace{p\mathbb{P}\left(\|\mathbf{X}_{\mathcal{S}}\boldsymbol{\beta}_{\backslash i}^\natural\|_2 \geq \sqrt{2(n-h)}\|\boldsymbol{\beta}_{\backslash i}^\natural\|_2\right)}_{\zeta_1}
$$

$$
+ \underbrace{p\mathbb{P}\left(\sum_{\ell=1}^{n-h}\mathbf{X}_{\ell,i}\left\langle\mathbf{X}_{\ell,:}, \boldsymbol{\beta}_{\backslash i}^\natural\right\rangle \gtrsim \sqrt{(n-h)\log(mnp)}\|\boldsymbol{\beta}_{\backslash i}^\natural\|_2, \|\mathbf{X}_{\mathcal{S}}\boldsymbol{\beta}_{\backslash i}^\natural\|_2 < \sqrt{2(n-h)}\|\boldsymbol{\beta}_{\backslash i}^\natural\|_2\right)}_{\zeta_2}.
$$

Since $\mathbf{X}$ are with i.i.d Gaussian entries, we have each row in $\mathbf{X}_{\mathcal{S}}\boldsymbol{\beta}^{\natural}_{\backslash i}$ be a Gaussian random vector with zero mean and variance $\|\boldsymbol{\beta}^{\natural}_{\backslash i}\|_2^2$. Hence, we have $\|\mathbf{X}_{\mathcal{S}}\boldsymbol{\beta}^{\natural}_{\backslash i}\|_2^2/\|\boldsymbol{\beta}^{\natural}_{\backslash i}\|_2^2$ be a $\chi^2$ random variable with freedom $n-h$, which leads to

$$\zeta_1 \overset{\textcircled{1}}{\leq} p\exp\left(\frac{n-h}{2}\left(\log 2 - 1\right)\right) \leq pe^{-0.65(n-h)},$$

where $\textcircled{1}$ is due to Lemma 15.

For $\zeta_2$, we notice that $\mathbf{X}_{\ell,i}$ is independent of the inner product $\langle\mathbf{X}_{\ell,:},\boldsymbol{\beta}^{\natural}_{\backslash i}\rangle$. Hence, we can view the product $\sum_{\ell=1}^{n-h}\mathbf{X}_{\ell,i}\langle\mathbf{X}_{\ell,:},\boldsymbol{\beta}^{\natural}_{\backslash i}\rangle$ as a Gaussian random variable $\mathsf{N}(0,\|\mathbf{X}_{\mathcal{S}}\boldsymbol{\beta}^{\natural}_{\backslash i}\|_2^2)$, which leads to

$$\zeta_2 \leq p\mathbb{P}\left(\sum_{\ell=1}^{n-h}\mathbf{X}_{\ell,i}\left\langle\mathbf{X}_{\ell,:},\boldsymbol{\beta}^{\natural}_{\backslash i}\right\rangle \gtrsim \sqrt{\log(mnp)}\|\mathbf{X}_{\mathcal{S}}\boldsymbol{\beta}^{\natural}_{\backslash i}\|_2\right) \overset{\textcircled{2}}{\leq} 2n^{-c}m^{-c}p^{-c},$$

where in $\textcircled{2}$ we use the tail bound of Gaussian random vectors. $\qquad\square$

**Lemma 9.** *For a fixed $\boldsymbol{\beta}\in\mathbb{R}^p$, we have $\mathbb{P}(\mathcal{E}_3(\boldsymbol{\beta})) \geq 1 - c_0 n^{-c_1}m^{-c_2}p^{-c_3}$.*

*Proof.* According to Lemma 16, we can decompose the index set $\{\ell: \ell \neq \pi^{\natural}(\ell)\}$ into three disjoint sets $\mathcal{I}_j$ such that $(i)$ indices $\ell$ and $\pi^{\natural}(\ell)$ do not fall into the same set $\mathcal{I}_j$; and $(ii)$ the cardinality of each set satisfies $h_j \triangleq |\mathcal{I}_j| \geq \lfloor\frac{h}{3}\rfloor$, $(1 \leq j \leq 3)$.

Then we can decompose product $\sum_{\ell\neq\pi^{\natural}(\ell)}\mathbf{X}_{\ell,i}\langle\mathbf{X}_{\pi^{\natural}(\ell),:},\boldsymbol{\beta}\rangle$ as

$$\sum_{\ell\neq\pi^{\natural}(\ell)}\mathbf{X}_{\ell,i}\left\langle\mathbf{X}_{\pi^{\natural}(\ell),:},\boldsymbol{\beta}\right\rangle = \sum_{j=1}^{3}\sum_{\ell\in\mathcal{I}_j}\mathbf{X}_{\ell,i}\left\langle\mathbf{X}_{\pi^{\natural}(\ell),:},\boldsymbol{\beta}^{\natural}\right\rangle.$$

With the union bound, we have

$$\mathbb{P}(\overline{\mathcal{E}}_3(\boldsymbol{\beta})) \leq p\cdot\sum_{j=1}^{3}\mathbb{P}\left(\sum_{\ell\in\mathcal{I}_j}\mathbf{X}_{\ell,i}\left\langle\mathbf{X}_{\pi^{\natural}(\ell),:},\boldsymbol{\beta}^{\natural}\right\rangle \gtrsim (\log mp)\sqrt{h_j}\|\boldsymbol{\beta}^{\natural}\|_2\right). \qquad(19)$$

Due to the properties of $\mathcal{I}_j$, we have $\mathbf{X}_{\ell,i}$ and $\langle\mathbf{X}_{\pi^{\natural}(\ell),:},\boldsymbol{\beta}^{\natural}\rangle$ be independent and hence

$$\mathbb{P}\left(\sum_{\ell\in\mathcal{I}_j}\mathbf{X}_{\ell,i}\left\langle\mathbf{X}_{\pi^{\natural}(\ell),:},\boldsymbol{\beta}^{\natural}\right\rangle \gtrsim \log(mnp)\sqrt{h_j}\|\boldsymbol{\beta}^{\natural}\|_2\right)$$

$$= \mathbb{P}\left(\left\|\mathbf{X}_{\ell\in\mathcal{I}_j,:}\boldsymbol{\beta}^{\natural}\right\|_2^2 \gtrsim \log(mnp)|h_j|\|\boldsymbol{\beta}^{\natural}\|_2^2\right)$$

$$+ \mathbb{P}\left(\left\|\mathbf{X}_{\ell\in\mathcal{I}_j,:}\boldsymbol{\beta}^{\natural}\right\|_2^2 \lesssim \log(mnp)|h_j|\|\boldsymbol{\beta}^{\natural}\|_2^2, \sum_{\ell\in\mathcal{I}_j}\mathbf{X}_{\ell,i}\left\langle\mathbf{X}_{\pi^{\natural}(\ell),:},\boldsymbol{\beta}^{\natural}\right\rangle \gtrsim \log(mnp)\sqrt{h_j}\|\boldsymbol{\beta}^{\natural}\|_2\right)$$

$$\overset{\textcircled{1}}{\leq} \exp\left[\frac{h_j}{2}\left(\log(c_0\log mnp) - c_0\log mnp + 1\right)\right] + \mathbb{P}\left(\sum_{\ell\in\mathcal{I}_j}\mathbf{X}_{\ell,i}\left\langle\mathbf{X}_{\pi^{\natural}(\ell),:},\boldsymbol{\beta}^{\natural}\right\rangle \gtrsim \sqrt{\log mnp}\|\mathbf{X}_{\ell\in\mathcal{I}_j,:}\boldsymbol{\beta}^{\natural}\|_2\right)$$

$$\overset{\textcircled{2}}{\lesssim} n^{-c_0}m^{-c_0}p^{-c_0} + n^{-c_1}m^{-c_1}p^{-c_1} \asymp n^{-c}m^{-c}p^{-c},$$

where $\textcircled{1}$ is due to Lemma 15 and $\textcircled{2}$ is due to the Gaussian tail bound. Plugging it in (19) then completes the proof. $\qquad\square$

**Lemma 10.** *We have* $\mathbb{P}(\mathcal{E}_4) \geq 1 - cn^{-c_0}m^{-c}p^{-c}$.

*Proof.* We complete the proof as

$$
\begin{aligned}
\mathbb{P}(\overline{\mathcal{E}}_4) &\overset{\textcircled{1}}{\leq} mp \cdot \mathbb{P}\left(\sum_{\ell=1}^{n} \mathbf{X}_{\ell,i}^{\top}\mathbf{W}_{\ell,j} \gtrsim \sigma\sqrt{n}\log(mnp)\right) \\
&\overset{\textcircled{2}}{\leq} mp\left[\mathbb{P}\left(\|\mathbf{X}_{:,i}\|_2 \gtrsim \sqrt{n\log(mnp)}\right) + \mathbb{P}\left(\sum_{\ell=1}^{n}\mathbf{X}_{\ell,i}^{\top}\mathbf{W}_{\ell,j} \gtrsim \sigma\sqrt{n}\log(mnp), \|\mathbf{X}_{:,i}\|_2 \lesssim \sqrt{n\log(mnp)}\right)\right] \\
&\overset{\textcircled{3}}{\leq} mp\left[\exp\left(\frac{n}{2}\left(\log(c_0\log(mnp)) - c_0\log(mnp) + 1\right)\right) + \mathbb{P}\left(\sum_{\ell=1}^{n}\mathbf{X}_{\ell,i}^{\top}\mathbf{W}_{\ell,j} \gtrsim \sigma\sqrt{\log mnp}\|\mathbf{X}_{:,i}\|_2\right)\right] \\
&\overset{\textcircled{4}}{\leq} mp\left(n^{-c_0}m^{-c_0}p^{-c_0} + n^{-c_1}m^{-c_1}p^{-c_1}\right) \asymp n^{-c_0}m^{-c}p^{-c},
\end{aligned}
$$

where $\textcircled{1}$ and $\textcircled{2}$ are due to the union bound, $\textcircled{3}$ is due to Lemma 15, and $\textcircled{4}$ is due to the tail bound for Gaussian random variable. $\qquad\square$

**Lemma 11.** *Conditional on the intersection of events* $\mathcal{E}_1 \bigcap \mathcal{E}_2(\boldsymbol{\beta}^\natural) \bigcap \mathcal{E}_3(\boldsymbol{\beta}^\natural) \bigcap \mathcal{E}_4$, *we have*

$$
|\boldsymbol{\beta}_{\mathsf{imax}}^\natural| \gtrsim |\boldsymbol{\beta}_{\mathsf{max}}^\natural| - \frac{\sqrt{\log(mnp)}}{\sqrt{n}}\|\boldsymbol{\beta}^\natural\|_2 - \frac{\sigma\log(mnp)}{\sqrt{n}},
$$

*where* $\mathsf{imax}$ *and* $\mathsf{max}$ *are defined as the indices of* $\widehat{\boldsymbol{\beta}}$ *and* $\boldsymbol{\beta}^\natural$ *with the largest magnitude, i.e.,* $\mathsf{imax} \triangleq \mathrm{argmax}_i|\widehat{\boldsymbol{\beta}}_i|$ *and* $\mathsf{max} \triangleq \mathrm{argmax}_i|\boldsymbol{\beta}_i^\natural|$, *respectively.*

*Proof.* To begin with, we define $\zeta_1^{(i)}$, $\zeta_2^{(i)}$, and $\zeta_3^{(i)}$ as

$$
\begin{aligned}
\zeta_1^{(i)} &= \frac{1}{n-h}\left(\sum_{\ell=1}^{n-h}\mathbf{X}_{\ell,i}^2\right)\boldsymbol{\beta}_i^\natural; \\
\zeta_2^{(i)} &= \frac{1}{n-h}\sum_{\ell=1}^{n-h}\mathbf{X}_{\ell,i}\left\langle\mathbf{X}_{\ell,:}, \boldsymbol{\beta}_{\backslash i}^\natural\right\rangle; \\
\zeta_3^{(i)} &= \frac{1}{n-h}\sum_{\ell=n-h+1}^{n}\mathbf{X}_{\ell,i}\left\langle\mathbf{X}_{\pi^\natural(\ell),:}, \boldsymbol{\beta}^\natural\right\rangle; \\
\zeta_4^{(i)} &= \frac{1}{n-h}\sum_{\ell=1}^{n}\mathbf{X}_{\ell,i}^{\top}\mathbf{W}_{\ell,j},
\end{aligned}
$$

respectively. Then we can write $\widehat{\boldsymbol{\beta}}_i$ as $\sum_{j=1}^{4}\zeta_j^{(i)}$. Due to the definition of $\mathsf{imax}$, we conclude $|\widehat{\boldsymbol{\beta}}_{\mathsf{imax}}| \geq |\widehat{\boldsymbol{\beta}}_{\mathsf{max}}|$, where $\mathsf{max}$ is defined as the index of $\boldsymbol{\beta}^\natural$ with the largest magnitude, i.e., $\mathsf{max} \triangleq \mathrm{argmax}_i|\boldsymbol{\beta}_i^\natural|$. With triangle inequality, we obtain

$$
\sum_{j=1}^{4}|\zeta_j^{(\mathsf{imax})}| \geq |\widehat{\boldsymbol{\beta}}_{\mathsf{imax}}| \geq |\widehat{\boldsymbol{\beta}}_{\mathsf{max}}| \geq |\zeta_1^{(\mathsf{max})}| - \sum_{j=2}^{4}|\zeta_j^{(\mathsf{max})}|. \tag{20}
$$

The following context separately discusses each term. First, we consider $|\zeta_1^{(\mathsf{imax})}|$ and $|\zeta_1^{(\mathsf{max})}|$. Conditional on $\mathcal{E}_1$, we have

$$
\begin{aligned}
|\zeta_1^{(\mathsf{imax})}| &\leq \left(1 + c_0\sqrt{\frac{\log(np)}{n-h}}\right)|\boldsymbol{\beta}_{\mathsf{imax}}^\natural| \lesssim |\boldsymbol{\beta}_{\mathsf{imax}}^\natural|; \\
|\zeta_1^{(\mathsf{max})}| &\geq \left(1 - c_0\sqrt{\frac{\log(np)}{n-h}}\right)|\boldsymbol{\beta}_{\mathsf{max}}^\natural| \gtrsim |\boldsymbol{\beta}_{\mathsf{max}}^\natural|. \tag{21}
\end{aligned}
$$

Then we turn to study the rest of terms. Conditional on $\mathcal{E}_2(\boldsymbol{\beta}^\natural) \bigcap \mathcal{E}_3(\boldsymbol{\beta}^\natural) \bigcap \mathcal{E}_4$, we have

$$|\zeta_2^{(\text{imax})}|, |\zeta_2^{(\text{max})}| \lesssim \sqrt{\frac{\log(mnp)}{n-h}} \|\boldsymbol{\beta}^\natural\|_2;$$

$$|\zeta_3^{(\text{imax})}|, |\zeta_3^{(\text{max})}| \lesssim \frac{\log(mnp)\sqrt{h}}{n-h} \|\boldsymbol{\beta}^\natural\|_2;$$

$$|\zeta_4^{(\text{imax})}|, |\zeta_4^{(\text{max})}| \lesssim \frac{\sigma\sqrt{n}\log(mnp)}{n-h}. \tag{22}$$

Combining (20), (21), and (22) then yields the lower-bound for $|\boldsymbol{\beta}_{\text{imax}}^\natural|$

$$|\boldsymbol{\beta}_{\text{imax}}^\natural| \gtrsim |\zeta_1^{(\text{imax})}| \geq |\zeta_1^{(\text{max})}| - \sum_{j=2}^{4} \left( |\zeta_j^{(\text{imax})}| + |\zeta_j^{(\text{max})}| \right)$$

$$\gtrsim |\boldsymbol{\beta}_{\text{max}}^\natural| - c_2 \sqrt{\frac{\log(mnp)}{n-h}} \|\boldsymbol{\beta}^\natural\|_2 - \frac{c_3 \log(mnp)\sqrt{h}}{n-h} \|\boldsymbol{\beta}^\natural\|_2 - \frac{c_4 \sigma\sqrt{n}\log(mnp)}{n-h},$$

which concludes the proof as $n \gtrsim h$.

$\square$

**Lemma 12.** *Conditional on the intersection of events $\mathcal{E}_1 \bigcap \mathcal{E}_2(\boldsymbol{\beta}^\natural) \bigcap \mathcal{E}_3(\boldsymbol{\beta}^\natural) \bigcap \mathcal{E}_4$, we have*

$$\|\widehat{\boldsymbol{\beta}} - \boldsymbol{\beta}^\natural\|_\infty \lesssim \frac{\log(mnp)}{\sqrt{n}} \|\boldsymbol{\beta}^\natural\|_2 + \frac{\sigma \log(mnp)}{\sqrt{n}}.$$

*Proof.* For an arbitrary index $i$, we consider the difference $\widehat{\boldsymbol{\beta}}_i - \boldsymbol{\beta}_i^\natural$, which can be written as

$$\widehat{\boldsymbol{\beta}}_i - \boldsymbol{\beta}_i^\natural = \underbrace{\frac{1}{n-h} \sum_{\ell=\pi^\natural(i)} \left(\mathbf{X}_{\ell,i}^2 - 1\right) |\boldsymbol{\beta}_i^\natural|}_{\zeta_1} + \underbrace{\frac{1}{n-h} \sum_{\ell=\pi^\natural(\ell)} \mathbf{X}_{\ell,i} \left\langle \mathbf{X}_{\ell,:}, \boldsymbol{\beta}_{\backslash i}^\natural \right\rangle}_{\zeta_2}$$

$$+ \underbrace{\frac{1}{n-h} \sum_{\ell=n-h+1}^{n} \mathbf{X}_{\ell,i} \left\langle \mathbf{X}_{\pi^\natural(\ell),:}, \boldsymbol{\beta}^\natural \right\rangle}_{\zeta_3} + \underbrace{\frac{1}{n-h} \sum_{\ell=1}^{n} \mathbf{X}_{\ell,i}^\top \mathbf{W}_{\ell,j}}_{\zeta_4}.$$

Conditional on the intersection of events $\mathcal{E}_1 \bigcap \mathcal{E}_2(\boldsymbol{\beta}^\natural) \bigcap \mathcal{E}_3(\boldsymbol{\beta}^\natural) \bigcap \mathcal{E}_4$, we can bound $\vartheta_i$ ($1 \leq i \leq 4$), as

$$|\zeta_1| \lesssim \sqrt{\frac{\log(np)}{n-h}} |\boldsymbol{\beta}_i^\natural|;$$

$$|\zeta_2| \lesssim \sqrt{\frac{\log(mnp)}{n-h}} \|\boldsymbol{\beta}_{\backslash i}^\natural\|_2;$$

$$|\zeta_3| \lesssim \frac{\log(mnp)\sqrt{h}}{n-h} \|\boldsymbol{\beta}^\natural\|_2;$$

$$|\zeta_4| \lesssim \frac{\sigma\sqrt{n}\log(mnp)}{n-h},$$

respectively, and complete the proof as $|\zeta_1| + |\zeta_2| \lesssim \sqrt{\frac{\log(mnp)}{n-h}} \|\boldsymbol{\beta}^\natural\|_2$ and $h \leq n$. $\square$

**Lemma 13.** *Conditional on the intersection of events $\mathcal{E}_1 \bigcap_{\ell=1}^{m} \mathcal{E}_2(\mathbf{B}_{\ell,:}^\natural) \bigcap_{\ell=1}^{m} \mathcal{E}_3(\mathbf{B}_{\ell,:}^\natural) \bigcap \mathcal{E}_4$, we have*

$$\left\langle \widehat{\mathbf{B}}_{\text{imax}}^{(\pi^\natural(i),j)}, \mathbf{B}^\natural \right\rangle \geq \frac{\left\|\|\mathbf{B}^\natural\|\right\|_F^2}{k} - \frac{m\sigma^2 \left(\log mnp\right)^2}{n} - \frac{\sqrt{m}\sigma \log(mnp)}{\sqrt{n}} \left\|\|\mathbf{B}^\natural\|\right\|_F,$$

*provided $n \gg k(\log mnp)^2$.*

*Proof.* First, we pick one arbitrary column $\boldsymbol{\beta}^\natural$ of $\mathbf{B}^\natural$. W.l.o.g. we assume that $\boldsymbol{\beta}^\natural_{\text{imax}} \geq 0$. Then we obtain

$$\widehat{\boldsymbol{\beta}}_{\text{imax}} \geq \boldsymbol{\beta}^\natural_{\text{imax}} - \|\widehat{\boldsymbol{\beta}}^{(\pi^\natural(i),j)} - \boldsymbol{\beta}^\natural\|_\infty,$$

where $\widehat{\boldsymbol{\beta}}^{(\pi^\natural(i),j)}$ denotes the corresponding column in $\widehat{\mathbf{B}}^{(\pi^\natural(i),j)}$. Then we obtain the following lower bound on $\widehat{\boldsymbol{\beta}}_{\text{imax}}\boldsymbol{\beta}^\natural_{\text{imax}}$

$$\widehat{\boldsymbol{\beta}}_{\text{imax}}\boldsymbol{\beta}^\natural_{\text{imax}} \geq (\boldsymbol{\beta}^\natural_{\text{imax}})^2 - |\boldsymbol{\beta}^\natural_{\text{imax}}|\|\widehat{\boldsymbol{\beta}}^{(\pi^\natural(i),j)} - \boldsymbol{\beta}^\natural\|_\infty. \tag{23}$$

Similarly, we can show (23) holds as well when $\boldsymbol{\beta}^\natural_{\text{imax}} < 0$. Recalling the definition of $\boldsymbol{\beta}^\natural_{\max}$, to put more specifically $|\boldsymbol{\beta}^\natural_{\max}| \geq |\boldsymbol{\beta}^\natural_i|$ ($1 \leq i \leq p$), we can further lower bound $\widehat{\boldsymbol{\beta}}_{\text{imax}}\boldsymbol{\beta}^\natural_{\text{imax}}$ by $(\boldsymbol{\beta}^\natural_{\text{imax}})^2 - |\boldsymbol{\beta}^\natural_{\max}|\|\widehat{\boldsymbol{\beta}}^{(\pi^\natural(i),j)} - \boldsymbol{\beta}^\natural\|_\infty$.

For $|\boldsymbol{\beta}^\natural_{\text{imax}}|$, we can lower bounded it by Lemma 11. While for $\|\widehat{\boldsymbol{\beta}}^{(\pi^\natural(i),j)} - \boldsymbol{\beta}^\natural\|_\infty$, we cannot directly use Lemma 12 since, strictly speaking, it concerns $\mathbf{X}$ with rows $\mathbf{X}_{\pi^\natural(i),:}, \mathbf{X}_{j,:}$ rather than $\widetilde{\mathbf{X}}_{\pi^\natural(i),:}, \widetilde{\mathbf{X}}_{j,:}$ However since they follow the same distributions, we can follow the same procedure and show

$$\|\widehat{\boldsymbol{\beta}}^{(\pi^\natural(i),j)} - \boldsymbol{\beta}^\natural\|_\infty \lesssim \frac{\log(mnp)}{\sqrt{n}}\|\boldsymbol{\beta}^\natural\|_2 + \frac{\sigma\log(mnp)}{\sqrt{n}}.$$

Then we obtain

$$\widehat{\boldsymbol{\beta}}_{\text{imax}}\boldsymbol{\beta}^\natural_{\text{imax}} \gtrsim \left(|\boldsymbol{\beta}^\natural_{\max}| - \frac{\sqrt{\log(mnp)}}{\sqrt{n}}\|\boldsymbol{\beta}^\natural\|_2 - \frac{\sigma\log(mnp)}{\sqrt{n}}\right)^2$$
$$- |\boldsymbol{\beta}^\natural_{\max}|\left(\frac{\log(mnp)}{\sqrt{n}}\|\boldsymbol{\beta}^\natural\|_2 + \frac{\sigma\log(mnp)}{\sqrt{n}}\right)$$
$$\overset{①}{\gtrsim} |\boldsymbol{\beta}^\natural_{\max}|^2 - |\boldsymbol{\beta}^\natural_{\max}|\left(\frac{\log(mnp)}{\sqrt{n}}\|\boldsymbol{\beta}^\natural\|_2 + \frac{\sigma\log(mnp)}{\sqrt{n}}\right)$$
$$- \frac{\log(mnp)}{n}\left(\|\boldsymbol{\beta}^\natural\|_2^2 + \sigma^2 \cdot \log(mnp)\right),$$

where in ① we use the relation $(a - b)^2 \geq a^2/2 - b^2$. Under the assumption $n \geq k(\log mnp)^2$, we have $|\boldsymbol{\beta}^\natural_{\max}| \geq \|\boldsymbol{\beta}^\natural\|_2/\sqrt{k} \gg \log(mnp)\|\boldsymbol{\beta}^\natural\|_2/(2\cdot\sqrt{n})$ and thus

$$|\boldsymbol{\beta}^\natural_{\max}|^2 - |\boldsymbol{\beta}^\natural_{\max}| \cdot \frac{\log(mnp)}{\sqrt{n}}\|\boldsymbol{\beta}^\natural\|_2 \overset{②}{\gtrsim} \left(\frac{1}{k} - \frac{\log(mnp)}{\sqrt{nk}}\right)\|\boldsymbol{\beta}^\natural\|_2^2,$$

where in ② we use the fact that $x^2 - 2ax$ is monotonically increasing in the region $[a, \infty)$; and the equality is achieved when $|\boldsymbol{\beta}^\natural_{\max}| = \|\boldsymbol{\beta}^\natural\|_2/\sqrt{k}$. Hence, we obtain

$$\widehat{\boldsymbol{\beta}}_{\text{imax}}\boldsymbol{\beta}^\natural_{\text{imax}} \gtrsim \left(\frac{1}{k} - \frac{\log(mnp)}{\sqrt{nk}} - \frac{\log(mnp)}{n}\right)\|\boldsymbol{\beta}^\natural\|_2^2 - \frac{\sigma\log(mnp)}{\sqrt{n}}\|\boldsymbol{\beta}^\natural\|_2 - \frac{\sigma^2(\log mnp)^2}{n}.$$

Having obtained the lower bound for one single column of $\mathbf{B}^\natural$, we complete the proof as

$$\langle\widehat{\mathbf{B}}^{(\pi^\natural(i),j)}_{\text{imax}}, \mathbf{B}^\natural\rangle = \sum_{\boldsymbol{\beta}^\natural} \widehat{\boldsymbol{\beta}}_{\text{imax}}\boldsymbol{\beta}^\natural_{\text{imax}}$$
$$\overset{③}{\gtrsim} \underbrace{\left(\frac{1}{k} - \frac{\log(mnp)}{\sqrt{nk}} - \frac{\log(mnp)}{n}\right)}_{\asymp k^{-1} \text{ since } n \gg k\log^2(mnp)}\||\mathbf{B}^\natural\||_F^2 - \frac{\sqrt{m}\sigma\log(mnp)}{\sqrt{n}}\||\mathbf{B}^\natural\||_F - \frac{m\sigma^2(\log mnp)^2}{n}$$

where ③ comes from a reorganization of the terms and the inequality $\sum_{\boldsymbol{\beta}^\natural}\|\boldsymbol{\beta}^\natural\|_2 \leq \sqrt{m}\||\mathbf{B}^\natural\||_F$. $\square$

**Lemma 14.** *We have $\mathbb{P}(\mathcal{E}_5) \geq 1 - 4n^{-c}p^{-c} - 2n^{-c'}m^{-c'}$.*

*Proof.* We begin the proof with the union bound

$$
\mathbb{P}\left(\overline{\mathcal{E}}_5\right) \leq \underbrace{\mathbb{P}\left(|x_i| \gtrsim \sqrt{\log np}, \; \exists\, 1 \leq i \leq p\right)}_{\leq\, 2np \cdot n^{-c}p^{-c} = 2n^{-c'}p^{-c'}} + \underbrace{\mathbb{P}\left(|\langle \boldsymbol{x}, \boldsymbol{\beta}^\natural\rangle| \gtrsim \sqrt{\log np}\|\boldsymbol{\beta}^\natural\|_2\right)}_{\leq\, \leq 2n \cdot n^{-c}p^{-c} = 2n^{-c'}p^{-c''}}
$$

$$
+ \underbrace{\mathbb{P}\left(|\boldsymbol{w}_i| \gtrsim \sigma\sqrt{\log nm}, \; 1 \leq i \leq m\right)}_{\leq\, 2mn \cdot n^{-c}m^{-c} = 2n^{-c'}m^{-c'}}
$$

$$
+ \underbrace{\mathbb{P}\left(\overline{\mathcal{E}}_5, \; \|\boldsymbol{x}\|_\infty \lesssim \sqrt{\log np}, \; |\langle \boldsymbol{x}, \boldsymbol{\beta}^\natural\rangle| \lesssim \sqrt{\log np}\|\boldsymbol{\beta}^\natural\|_2, \; \|\boldsymbol{w}\|_\infty \lesssim \sigma\sqrt{\log nm}\right)}_{\triangleq\,\vartheta},
$$

where $\boldsymbol{w}_{(\cdot)}$ and $\widetilde{\boldsymbol{w}}_{(\cdot)}$ denote the corresponding entries from $\mathbf{W}$ and $\widetilde{\mathbf{W}}$.

Then we would prove that $\vartheta$ is zero. The technical details are attached in the following. Due to the fact that $\widehat{\mathbf{B}}_{\mathsf{imax}}^{(i,j)}$ shares the same support set as $\widehat{\mathbf{B}}_{\mathsf{imax}}$, we have

$$
\left\|\left(\widehat{\mathbf{B}}_{\mathsf{imax}} - \widehat{\mathbf{B}}_{\mathsf{imax}}^{(i,j)}\right)^\top \boldsymbol{x}\right\|_2^2 = \sum_{\boldsymbol{\beta}^\natural}\left[\boldsymbol{x}_{\mathsf{imax}}\left(\widehat{\boldsymbol{\beta}}_{\mathsf{imax}} - \widehat{\boldsymbol{\beta}}_{\mathsf{imax}}^{\backslash(i,j)}\right)\right]^2 \overset{\textcircled{1}}{\lesssim} (\log np)\sum_{\boldsymbol{\beta}^\natural}\left\|\widehat{\boldsymbol{\beta}}_{\mathsf{imax}} - \widehat{\boldsymbol{\beta}}_{\mathsf{imax}}^{\backslash(i,j)}\right\|_2^2
$$

$$
= (\log np)\sum_{\boldsymbol{\beta}^\natural}\left\|\widehat{\boldsymbol{\beta}}_{\mathsf{imax}} - \widehat{\boldsymbol{\beta}}_{\mathsf{imax}}^{\backslash(i,j)}\right\|_\infty^2,
$$

where in $\textcircled{1}$ we condition on the relation $\|\boldsymbol{x}\|_\infty \lesssim \sqrt{\log np}$.

Regarding $\|\widehat{\boldsymbol{\beta}}_{\mathsf{imax}} - \widehat{\boldsymbol{\beta}}_{\mathsf{imax}}^{\backslash(i,j)}\|_\infty$, we have $\|\widehat{\boldsymbol{\beta}}_{\mathsf{imax}} - \widehat{\boldsymbol{\beta}}_{\mathsf{imax}}^{\backslash(i,j)}\|_\infty \leq \|\widehat{\boldsymbol{\beta}} - \widehat{\boldsymbol{\beta}}^{\backslash(i,j)}\|_\infty$. Again, this fact relies heavily on the fact such that $\widehat{\mathbf{B}}_{\mathsf{imax}}^{(i,j)}$ shares the same support set as $\widehat{\mathbf{B}}_{\mathsf{imax}}$. Otherwise, the best result we can have is $\|\widehat{\boldsymbol{\beta}}_{\mathsf{imax}} - \widehat{\boldsymbol{\beta}}_{\mathsf{imax}}^{\backslash(i,j)}\|_\infty \leq \max\left(\|\widehat{\boldsymbol{\beta}}_{\mathsf{imax}}\|_\infty, \|\widehat{\boldsymbol{\beta}}_{\mathsf{imax}}^{\backslash(i,j)}\|_\infty\right)$. Afterwards, we obtain

$$
\|\widehat{\boldsymbol{\beta}}_{\mathsf{imax}} - \widehat{\boldsymbol{\beta}}_{\mathsf{imax}}^{\backslash(\pi^\natural(i),j)}\|_\infty \leq (n-h)^{-1}\left\|\left(\mathbf{X}_{i,:}\mathbf{X}_{i,:}^\top + \mathbf{X}_{j,:}\mathbf{X}_{j,:}^\top - \widetilde{\mathbf{X}}_{i,:}\widetilde{\mathbf{X}}_{i,:}^\top - \widetilde{\mathbf{X}}_{j,:}\widetilde{\mathbf{X}}_{j,:}^\top\right)\boldsymbol{\beta}^\natural\right\|_\infty
$$

$$
+ (n-h)^{-1}\left\|\mathbf{X}_{\pi^\natural(i),:}\boldsymbol{w}_{\pi^\natural(i)} + \mathbf{X}_{i,:}\boldsymbol{w}_i - \widetilde{\mathbf{X}}_{\pi^\natural(i),:}\widetilde{\boldsymbol{w}}_{\pi^\natural(i)} - \widetilde{\mathbf{X}}_{i,:}\widetilde{\boldsymbol{w}}_i\right\|_\infty
$$

$$
\lesssim (n-h)^{-1}\|\boldsymbol{x}\langle\boldsymbol{x},\boldsymbol{\beta}^\natural\rangle\|_\infty + (n-h)^{-1}\|\boldsymbol{x}\boldsymbol{w}\|_\infty
$$

$$
\lesssim (n-h)^{-1}\|\boldsymbol{x}\|_\infty|\langle\boldsymbol{x},\boldsymbol{\beta}^\natural\rangle| + (n-h)^{-1}\|\boldsymbol{x}\|_\infty\|\boldsymbol{w}\|_\infty
$$

$$
\overset{\textcircled{2}}{\lesssim} (n-h)^{-1}(\log np)\|\boldsymbol{\beta}^\natural\|_2 + (n-h)^{-1}\sigma\sqrt{(\log np)(\log mn)},
$$

where in $\textcircled{2}$ we use the condition $\|\boldsymbol{x}\|_\infty \lesssim \sqrt{\log np}$, $|\langle\boldsymbol{x},\boldsymbol{\beta}^\natural\rangle| \lesssim \sqrt{\log np}\|\boldsymbol{\beta}^\natural\|_2$, and $\|\boldsymbol{w}\|_\infty \lesssim \sigma\sqrt{\log nm}$.

Iterating over all columns $\mathbf{B}^\natural$, we can show $\mathcal{E}_5$ holds with probability one provided $\|\boldsymbol{x}\|_\infty \lesssim \sqrt{\log np}$, $|\langle\boldsymbol{x},\boldsymbol{\beta}^\natural\rangle| \lesssim \sqrt{\log np}\|\boldsymbol{\beta}^\natural\|_2$, and $\|\boldsymbol{w}\|_\infty \lesssim \sigma\sqrt{\log nm}$, in other words, $\vartheta$ is zero. $\square$

## C  ADDITIONAL NUMERICAL EXPERIMENTS WITH REAL-WORLD DATA

This subsection evaluates our algorithm on the MNIST dataset (LeCun et al., 1998). We first create the sparse matrix $\mathbf{B}^{\natural}$ as a block-diagonal matrix $\text{diag}\left(\text{image}_0^{\text{train}}, \text{image}_0^{\text{test}}, \text{image}_1^{\text{train}}, \cdots \text{image}_9^{\text{test}}\right)$, where $\text{image}_i^{\text{train}} \in \mathbb{R}^{28 \times 28}$ (resp. $\text{image}_i^{\text{test}} \in \mathbb{R}^{28 \times 28}$) represents an arbitrary image of digit $i$ ($0 \leq i \leq 9$) in the training (resp. test) set. Then, we create a Gaussian sensing matrix $\mathbf{X}$ and a noise matrix $\mathbf{W}$. Afterwards, we permute the sensing results and apply our algorithm in Algorithm 1 to reconstruct the images. As the benchmark, we ignore the missing correspondence and directly estimate $\mathbf{B}^{\natural}$ with Lasso estimator. In other words, we let $\mathbf{\Pi}^{\text{opt}}$ in (3) be the identity matrix and use it to estimate the images. An illustration of the reconstructed images is put in Figure 6.

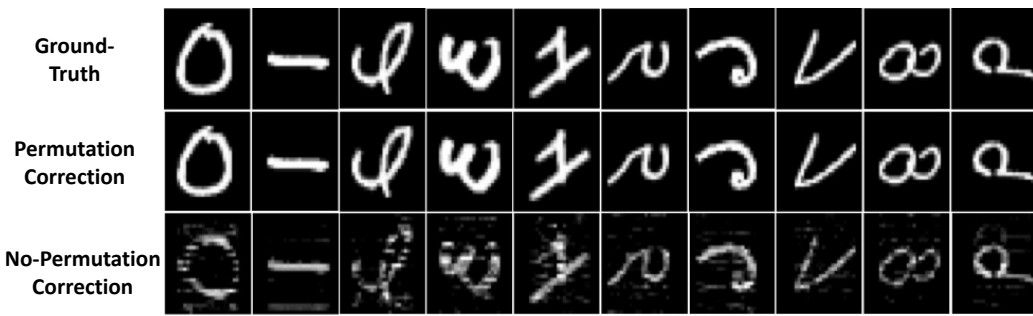

Figure 6: We set the sample number $n$ as 100 and the sensing noise variance $\sigma^2$ as 1. (**Top**) The ground-truth images. (**Middle**) The reconstructed images with our algorithm using both (2) and (3). (**Bottom**) The reconstructed images only using (3), where $\mathbf{\Pi}^{\text{opt}}$ is simply set as $\mathbf{I}$.

## D  USEFUL FACTS

This section collects some useful facts about probability for the sake of self-containing.

**Lemma 15** ((Dasgupta & Gupta, 2003)). *For a $\chi^2$-random variable $Z$, which is with freedom $\ell$, we conclude*

$$\mathbb{P}(Z \leq t) \leq \exp\left[\frac{\ell}{2}\left(\frac{t}{\ell} - \frac{t}{\ell} + 1\right)\right], \quad t < \ell;$$

$$\mathbb{P}(Z \geq t) \leq \exp\left[\frac{\ell}{2}\left(\frac{t}{\ell} - \frac{t}{\ell} + 1\right)\right], \quad t > \ell.$$

**Lemma 16** ((Pananjady et al., 2018)). *Suppose the permutation matrix $\mathbf{\Pi}$ with Hamming distance $h$ from the identity matrix $\mathbf{I}$, namely, $\mathsf{d}_{\mathsf{H}}(\mathbf{I}, \mathbf{\Pi}) = h$. We can decompose the index set $\{i : i \neq \pi(i)\}$ into 3 independent sets $\mathcal{I}_i$ ($1 \leq i \leq 3$) such that the cardinality of each set satisfies $|\mathcal{I}_i| \geq \lfloor h/3 \rfloor \geq h/5$.*

