# OpenReview forum: "One-Step Estimator for Permuted Sparse Recovery"
_ICLR.cc/2023/Conference — Submitted to ICLR 2023_

### Official Review · Reviewer_BXq7 · 2022-10-14

**Confidence:** 5
**Correctness:** 3
**Technical Novelty And Significance:** 3
**Empirical Novelty And Significance:** 2
**Recommendation:** 6

**Clarity, Quality, Novelty And Reproducibility:**

The paper is written clearly, while there are a few typos as mentioned above.

To the best of my knowledge, the result is novel, despite the fact that it is built upon the analysis framework of several prior works.

**Strength And Weaknesses:**

*STRENGTH*:
- The paper derives two solid theorems about permutation recovery:
  - Theorem 2 states that exact permutation recovery is impossible for
  fixed $m$ and $p$, as long as $n$ approaches infinity. The condition
  is very simple and the theorem is very clean.
  - Theorem 3 states that the ground-truth permutation can be recovered
  exactly with high probability by solving the linear assignment
  problem (2), as long as (iv) SNR is larger than some constant,
  (iii) the number of permuted rows is small enough, (ii) the stable
  rank of the ground-truth matrix is larger than some logarithm of the number
  of samples, and (i) we have sufficiently many samples. Once the
  permutation is correctly recovered, the recovery of the ground-truth
  matrix ensues, based on the results in compressed sensing.

- Figure 2 presents an interesting phenomenon where
  the number of samples is actually a double-edged quantity in this
  problem. As Remark 2 discussed, larger $n$ violates the stable rank
  condition (ii). Another perspective might be found in Theorem 2 and
  Theorem 3: With $m$ and $p$ fixed and $n$ approaching infinity, the
  recovery of the permutation becomes impossible. All these suggest
  that, in order for exact recovery, the parameters $m,p,n,h$ might
  have a more sophisticated dependency than the theorems reveal. It
  would be interesting future work to discover such dependency.

- The proof outline (Sec 4.2) is great as it helps the reader to better
  appreciate the technical challenge and thus the contribution of the
  paper.

- I found Table 1 informative. It positions the paper in the
  literature very well. From it, one can also find that the paper
  advances our theoretical understanding of the problem.

*WEAKNESS AND QUESTIONS*:
- In Theorem 2, the supremum is taken over $B_{p,m}$. Why this is
  required? I thought that, since $n$ approaches infinity, this set
  would eventually be the entire set of $p\times m$ matrices.
  Furthermore, should it be stated in Theorem 2 that the ground-truth
  matrix is given as in Theorem 1? Or this is actually not needed for
  the sparse case?
- Personally, I think that, if possible, it would be greater if an
  explanation is given regarding the phenomenon discussed in Remark 1
  and Figure 1. I guess the authors have an answer as to keeping more
  entries might break down the proof. This discussion is important as
  the situation is very different from what is commonly known.
- Sec 4.3 looks a little bit mysterious to me. It seems not very
  rigorous to claim that the proposed estimator is computationally
  optimal. For example, despite being incorrect, the estimator that
  returns an identity permutation and a zero matrix is computationally
  optimal, as it does not require any computation, let alone matrix
  multiplication which is needed by the proposed method. The
  point is, to make such a claim, one needs to prove that (informally
  speaking) there is no estimator that takes fewer operations
  (constrained to its correctness).
- Why is the paper titled *one-step estimator*? The proposed algorithm takes
  two steps, first recovering the permutation and then the sparse matrix.

*MINOR COMMENTS*:
- Typos or grammar problems at:
  - the first line of page 5 (sine -> since)
  - footnote 4 (extend -> extent)
  - the line below Equation 5 (accomplish -> accomplished)
  - the last line of page 7 (is -> are)

- There seems to be some citation issue at the transition of page 4
  and page 5. Please fix it.

- In the abstract, it is claimed that the case of the sparse unknown
  matrix in the *insufficient samples regime* is considered for the
  first time. This might be an overclaim. See for example
  - [1] Compressed sensing with unknown sensor permutation (ICASSP
    2014). There, the *insufficient samples regime* is studied
    computationally (Sec 3).
  - [2] Sparse recovery with shuffled labels: statistical limits and
    practical estimators. There, the *insufficient samples regime* is
    studied theoretically and computationally (as the paper noted).
  - [3] Homomorphic sensing: sparsity and noise (ICML 2021). There, the
    *insufficient samples regime* is studied theoretically and
    computationally (Sec 3).



**Summary Of The Paper:**

The paper considers the problem of unlabeled sparse recovery under
multiple linear measurements. Statistical theorems are derived for
understanding the proposed estimator.

**Summary Of The Review:**

The paper provides some solid theoretical results, and reports and explains some intriguing phenomena for the problem of unlabeled permutation recovery. While I have a few questions and comments, they are somewhat minor and can be easily addressed. In summary, I think this is an interesting paper and I vote for accept. (I did not check the proof in the appendix.)

---

> ### Author Response · Authors · 2022-11-19
> **Thank you for the review**
>
>
> Thank you for your encouraging comments.
>
> **1. Why is the supremum is taken over $B_{p, m}$ since $n$ approaches to infinity?**
>
> Thanks. We clarify that we allow measurement number $m$ and variable length $p$ approach infinity as well.  This makes the $B_{p, m, n, k}$ only cover a subset of all matrices.
>
> **2. Whether $B^{\natural}$ is known in Theorem 2?**
>
> We do not assume the knowledge of $B^{\natural}$. This is the reason  why we have the term $\frac{m \log {p\choose k}}{n}$ in the right-hand side of the inequality in Theorem 2.
>
> **3. Adding explanation why only keeping one non-zero element in each column.**
>
> Thank you for the suggestion. We have added in the revision (Remark 2) the explanation. Basically, insufficient samples leads to a poor approximation of $B^{\natural}$ with $X^T Y$. To reduce the approximation error, we should threshold as many non-zero elements as possible.
>
>
> **4. Questions computational optimality.**
>
> The computational time includes the cost for estimating both $\Pi$ and $B$. The cost of estimating the sparse matrix $B$ is the dominating component as it involves Lasso. Thanks for the suggestion of simply using identity matrix as the reconstructed permutation matrix to estimate $B$. That would incur essentially similar computational cost because recovering $B$ dominates the computation.
>
> Instead of defining optimiality, the revised version focuses on presenting the computational complexity. Hopefully this would avoid confusion if any. Thank you.
>
> **5. Title for one-step estimator.**
>
> Thank you for the comment. We view the estimation for $\Pi^{\textup{opt}}$ and that for $B^{\textup{opt}}$ as a component of one step.  By "one-step", we mean to say that we only estimate $(\Pi^{\textup{opt}}, B^{\textup{opt}})$ once instead of take multiple rounds to iteratively refine the estimate for $(\Pi^{\textup{opt}}, B^{\textup{opt}})$.
>
> We guess "One-Stage estimator" might be even more clear, although we personally like "One-step" better than "One-Stage". We have added in the paper the (brief) explanation of the  "one-Step" estimator: _We propose a one-step estimator for the correspondence recovery, which consists of two sub-parts: one for $\Pi^{\natural}$ and another for $B^{\natural}$._
>
>
> Thank you.
>
>
> **6. Comparison with previous work.**
>
> We thank you for carefully surveying the related works. We are familiar with the existing works (which we have cited). These existing works focus on the singe measurement case, i.e., $m =1$. In this paper, we focus on the multiple measurement scenario, i.e. $m>1$, which is the first work (to the best of our knowledge). Note that single  measurement setting and multiple measurement setting have very different properties. For example, the single measurement requires the $\textup{SNR}$ to be at least $O(n^c)$ for the correct permutation recovery (c.f. Sparse recovery with shuffled labels: statistical limits and practical estimators). Meanwhile, we find that the minimum $\textup{SNR}$ can be reduced to constant with more measurements.

---

> ### Author Response · Authors · 2022-12-09
> **Feedback on the rebuttal**
>
> Dear Reviewer,
>
> Thank you again for your detailed comments. We hope our revision and the answers to your questions have addressed your concerns. Please let us know if we should answer any additional questions.  Thanks a lot.
>
> Sincerely,
>
> Authors

---

### Official Review · Reviewer_7u4U · 2022-10-25

**Confidence:** 2
**Correctness:** 3
**Technical Novelty And Significance:** 4
**Empirical Novelty And Significance:** 3
**Recommendation:** 6

**Clarity, Quality, Novelty And Reproducibility:**

As mentioned above, the clarity of writing can be improved. The results are novel and are clearly interesting. I was not able to check the proofs in detail.

**Strength And Weaknesses:**

Strengths:
To the best of my knowledge, the novel results go clearly beyond existing work (constant SNR in the undersampled regime) and should be interesting to people working on these topics.

Weaknesses:
1.The dependence of the number of measurements on the sparsity level is quadratic instead of linear.
2. Also, I don't really understand Section 4.2. Why does the Theorem follow form inequality (4)? I also don't understand why the inequality on page 8 is true. I am also not sure whether I understand how $()_{\imax}$ is defined.
3. In general, the presentation of the material can be improved; see some of the comments below.

Further comments:
1. page 3: It might be good to explain how the expression $ \log \det \left( I + \frac{B^T B}{\sigma^2}  \right) $ relates to the SNR!
2. page 3: "let $n$ approach infinity". Consider replacing this statement with something more rigorous. Is this an asymptotic or non-asymptotic statement?
3. page 4: It seems to me that $\mathcal{B}_{p,m}$ also depends on $k$. It would be good to make this dependence explicit, for example, by adding $k$ as a subscript.
4. page 5, Theorem 3 and Corollary: Consider replacing "Consider the large-system limit..." by something more rigorous. (It is not clear whether this is an asymptotic or non-asymptotic statement!)
5. page 5, Corollary 1: "where $n$ and $p$ are sufficiently large. Consider replacing this by a more rigorous statement.
6. page 5, Theorem 3: Has $h$ been already properly introduced? If yes, it might be a good idea to recall its definition!
7. page 7: Why does $ \Vert B^{\sharp} \Vert_F $ affect the reconstruction error? I do not understand this statement.
8. page 8, Discussion w.r.t. sparsity number $k$: In contrast to your experiments, Theorem 2 does not say that one needs to increase the SNR when one increases the sparsity level $k$. Please add an explanation why this is not a contradiction.

Typos:
1. page 5 sin(c)e all these works

**Summary Of The Paper:**

This work considers the problem of reconstructing sparse vectors from an underdetermined system of equations when the measurements are commuted.
First, it gives a lower bound on the required number of measurements given a certain SNR.
Then, it analyses an estimator which gives corresponding upper bounds (depending on the number of measurements, signal-noise-ratio, sparsity level, ambient dimension, etc.)
The paper shows that one can reconstruct the sparse signals, even when the noise level is constant and when the number of measurements is quadratic in the sparsity level.
Numerical experiments corroborate the theoretical findings.

**Summary Of The Review:**

The paper has novel and interesting results, but I feel that the presentation can be significantly improved. I would recommend (borderline) acceptance.

---

> ### Author Response · Authors · 2022-11-19
> **Thank you for the review**
>
> Dear Reviewer,  Thank you.
>
>
> **1. Quadratic dependence of sparsity level on the sample number.**
>
> Thank you for this comment. With minor modification of Lemma 13 (by using the complete expression instead of the simplified one from Lemma 11), this bound has been improved to be linear.
>
>
> **2. Questions regarding the proof outline.**
>
> We have revised the corresponding part per your suggestion; thank you. Here, we provide a brief summary:
>
> Inequality (4) comes from the requirement $\Pi^{\natural}$ is  the solution of Equation (2). The sufficient condition in inequality (5) requires that each row's inner product on the left hand side in inequality (4) is larger than its counterpart on the right hand side. As for the inequality on page 8, we now move it to stage II in page 7 and its detailed derivation is also added. For the notation $\hat{\beta}_{imax}$, it selects the maximum entry in $\hat{\beta}$, i.e.,
>
> $\hat{\beta}_{imax} = \textup{argmax}_i |\hat{\beta}_i|$.
>
> **3. Relation between $\log\det(I + B^T B/\sigma^2)$ and $\textup{SNR}$.**
>
> Their relation is discussed in Remark $1$. When $B$ is rank-1, $\log\det(I + B^T B/\sigma^2)$  is $\log(1 + m \cdot \textup{SNR})$. When $B$ is with full-rank and uniform eigenvalues, $\log\det(I + B^T B/\sigma^2)$ is written as $rank(B) \cdot \log(1 + \textup{SNR})$.
>
> **4. Question about the consistency between discussion w.r.t sparsity number $k$ on page 8 and Theorem 2.**
>
> Yes, they are consistent. When $k$ increases from $15$ to $20$, we have set $\mathcal{B}_{n, p, m, k}$ to cover a boarder class of matrices, which implies a higher $\textup{SNR}$ for correspondence recovery. This is consistent with our findings, i.e., a larger $k$ makes the permutation reconstruction harder.
>
> Also, thank you for other comments on notations etc. Your suggestions have been incorporated into the revision.

---

> ### Author Response · Authors · 2022-12-09
> **feedback on the rebuttal?**
>
> Dear Referee,
>
> Thanks again for the constructive suggestions. We hope our revision  and the rebuttal were able to address your concerns on (e.g.,) the quadratic dependence and some of the presentations. Please let us know if there are further questions we could answer to help clarify any standing concerns. Thank you.
>
> Sincerely,
>
> Authors

---

### Official Review · Reviewer_Q6i9 · 2022-10-25

**Confidence:** 3
**Correctness:** 4
**Technical Novelty And Significance:** 3
**Empirical Novelty And Significance:** 3
**Recommendation:** 6

**Clarity, Quality, Novelty And Reproducibility:**

Clarity: The paper is clearly written.
Quality: I think this is a good paper and vote to accept it.
Novelty: The paper has some technical novelty involved in extending and modifying prior proof techniques to the setting that they consider.
Reproducibility: As far as I could tell, the authors do not provide the code for their experiments. However, there is a detailed description of the method and so they are probably reproducible.

**Strength And Weaknesses:**

Strengths:
1. The paper provides new lower bounds on SNR and sample complexity required for recovery for the problem of unlabelled sparse recovery under multiple measurements.
2. It demonstrates a simple estimator which requires much lower SNR than prior works, in the regime where the number of samples is much smaller than the ambient dimension.



**Summary Of The Paper:**

This paper considers the problem of unlabelled sparse recovery under multiple measurements. More formally, Given $(Y, X)$ such that $Y = \Pi^* X B^*+ W$, where $W \in \mathbb R^{n \times m}$, $W_{i, j} \sim N(0, \sigma^2)$, $X \in \mathbb R^{n \times p}$, $X_{i, j} \sim N(0, 1)$ and each column of $B^* \in \mathbb R^{p \times m}$ is $k$-sparse, the goal is to recover $B^*$ and the unknown permutation matrix $\Pi^*$.

They show the following:

1. The sample complexity $n$ should be at least of the order $\Omega(k \log p)$ where $k$ is the sparsity of the signal vectors and $p$ is the ambient dimension. Also, the SNR is lower bounded by $\log det(I + ({B^*}{B^*}^T)/\sigma^2) \geq \log(n) + (m/n) \log(C(p,k))$.
2. By formulating the correspondence recovery as a linear assignment problem (LAP), they show that the correct permutation matrix
can be obtained when SNR is above certain positive constant.

Their results are the first ones s.t. SNR $\geq \tilde \Omega(1)$ is sufficient to obtain the correct permutation matrix with insufficient samples.

**Summary Of The Review:**

I think this is a good paper and vote to accept it. The paper seems to provide a concrete improvement over the prior work and the estimator is simple. I have not read the proof in the appendix.

---

> ### Author Response · Authors · 2022-11-19
> **Thank you for the review**
>
> Dear Reviewer:
>
> We highly appreciate your encouraging comments and suggestions. Indeed, the matlab implementation of our algorithm is  straightforward, as you mentioned.  In the supplementary material, the zip file contains all the matlab code needed to reproduce Figure 3 (upper-left plot), up to the simulation variations.
>
> Note that while the core algorithm only contains a few lines of matlab code, we need to resort to a solver for the assignment problem fin order to reproduce Figure 3. The C++ file is provided along the pre-compiled mex files for both windows and mac.
>
> In a matlab console, simply type
> >> run_n100p500k10
>
> to reproduce Figure 3 (upper left panel). Note that the number of simulations is set to be 100 in the code, while we used a much large number of simulations for Figure 3 in the paper.
>
> For the illustration, we pose the first few lies of the file 'run_n100p500k10.m',
>
> function run_n100p500k10
>
> % Compiled mex files for both windows and mac are already provided in the
>
> % folder. One can also un-comment the line below to re-compile the mex file.
>
> % mex -largeArrayDims auctionAlgorithmSparseMex.cpp -lut
>
> simul_param.max_case = 100;
>
> ...
>
> Please let us know if the matlab code runs properly for you. Thank you.

---

> ### Author Response · Authors · 2022-12-09
> **feedback on the rebuttal?**
>
> Dear Reviewer,
>
> Thank you again for the supportive review and constructive comments (e.g., suggesting us to provide the code). We hope the code we uploaded in the rebuttal could be useful for you to reproduce our results. Please let us know if there are further questions. Thank you.
>
> Sincerely,
>
> Authors.

---

### Official Review · Reviewer_KfQu · 2022-10-26

**Confidence:** 3
**Correctness:** 3
**Technical Novelty And Significance:** 3
**Empirical Novelty And Significance:** 2
**Recommendation:** 5

**Clarity, Quality, Novelty And Reproducibility:**

This paper is generally well-written. But there is a lack of rigorous/intuitive interpretations of the main theoretical results.

The technical results seem to be novel.

Although this paper only involves synthetic experiments, it is better to upload the code for reproducibility.

Some minor problems:

(a) Page 2: provide a reference for the linear assignment problem (LAP).

(b) Eq. (3): provide clear definition of $\||\mathbf{B}\||_1$.

Does it mean the sum of entries of $\mathbf{B}$ (in magnitude), or the operator norm $\||\mathbf{B}\||_{1\to 1}$?

(c)  Page 4: the definition of $\mathrm{supp}(\mathbf{B})$ should be given.

**Details Of Ethics Concerns:**

NA.

**Strength And Weaknesses:**

Strength:

The studied problem is at least of sufficient theoretical interest, and the technical results seem to be novel. Besides, the proposed algorithm is sufficiently simple.

Weaknesses:

1. The theoretical results are not clearly interpreted. For example, the authors mentioned, "For the sample number $n$, the lower bound requires $n$ to be at least of order $O(k \log p)$; while Corollary 1 requires $n$ to be  $k(\log n)(\log^2 m n p)$ " on page 6. Where does the lower bound come from? I guess it is not derived from Theorem 2 in this paper but from the lower bounds established in standard compressed sensing literature. The authors mentioned on page 2 that the lower bound is established with respect to both the number of samples and SNR. But I cannot see it clearly from the statement of Theorem 2. In addition, it is fairer to compare the lower bound with the upper bound for the number of samples established in Theorem 3 (which has a quadratic dependence on $k$), rather than Corollary 1, which is based on a seemingly restrictive assumption.

2. It is counter-intuitive to see that no matter how large $k$ is, the thresholding operator used in Eq. (2) that only keeps the element with the largest magnitude in each column works well. I hope that the authors can provide more intuitions/explanations for this, instead of only presenting Figure 1. The authors mentioned that "In the supplementary material, we give a rigorous explanation of this phenomenon" on page 5. But I cannot find where is a rigorous explanation. Please specify it.

3. The authors claimed in the conclusion that "we considered the unlabeled sparse recovery with multiple measurements for the first time". Although the studied problem is interesting in theory. From the current submission, I cannot see how practically meaningful it is. I hope that the authors can provide some numerical experiments on real data to back up the theory.

**Summary Of The Paper:**

In this paper, the authors provide theoretical guarantees for unlabeled sparse recovery with multiple measurements. They first establish the information-theoretic lower bounds with respect to the number of samples and signal-to-noise ratio (SNR) for the reconstruction of both the permutation and signal matrices. Then, they propose a one-step approach for reconstructing the permutation matrix (and the signal matrix can be easily reconstructed by using the estimated permutation matrix and standard compressed sensing techniques). Some experiments on synthetic data are provided to support the theoretical results.

**Summary Of The Review:**

Overall, I think that although this submission is interesting in theory, it is weak in both interpreting the main theoretical results and presenting the practical motivation. Therefore, I am inclined to rejection.

---

> ### Author Response · Authors · 2022-11-19
> **Thank you for the review**
>
> Dear Reviewer, thank you for your valuable comments.
>
>
> 1. The minimum sample number $n\geq c k\log p$ comes from the literature in compressive sensing, which is stated in Section 3.1.
>
> As for the quadratic dependence on $k$, we have refined our analysis and have improved it to linear dependence. The revision only takes place in Lemma 1, 11, and 13, with minor modifications.  Thank you again for pointing this out.
>
>
> 2. The intuition behind $\textup{thres}$, i.e., why we should only keep the element with the largest magnitude in $\textup{thres}(\cdot)$:. The intuition is that the insufficient samples  lead to a poor approximation of $B^{\natural}$ with $X^{T}Y$.
> Keeping more non-zero values in $\textup{thres}(\cdot)$ will risk decreasing the
> correlation $\langle \textup{thres}(X^T Y), B^{\natural}\rangle$.
>
> As for the formal explanation, it is given in Lemma 11 (e.g., Eq. (20)) and Lemma 14.  In Lemma 11, we have the lower bound of the  magnitude $|\widehat{\beta}_{\textup{imax}}|$ be guaranteed with the current $\textup{thres}(\cdot)$.
>
> In Lemma 14, we have the reconstruction error $\|(\hat{B}_{imax}^{(i, j)}- \hat{B})x\|_2$
>
> be controlled by $\sum_{\beta}|\hat{\beta}_{imax}^{\setminus (i, j)}-\hat{\beta}|$.
>
> Provided more non-zero elements are kept in $\textup{thres}(\cdot)$, these two values
>
> (lower bound on $|\hat{\beta}_{\textup{imax}}|$
>
> and  $\|(\widehat{B}_{imax}^{(i, j)}- \widehat{B})x\|_2$)
>
> will lose control, which is also verified by numerical experiments. The revised version should clarify the intuition and  specify the locations of the formal explanation.
>
> 3. Numerical experiments on real data.  We have evaluated our algorithm on the MNIST dataset. The results can be found in Appendix C. Thank you.
>
> 4. $\|B\|_1$ means the sum of entries of $B$ in magnitude.
>
> Again, thank you for your review and suggestions (including the suggestion of uploading the code), and please let us know if there are further questions.

---

> > ### Author Response · Authors · 2022-12-09
> > **feedbacks on the rebuttal?**
> >
> > Dear Referee,
> >
> > Thank you again for the comments and suggestions. We hope our revision/rebuttal has addressed your concerns on the quadratic dependence and on the real-data experiments. Please let us know if there are further questions we should try to answer. Thanks a lot.
> >
> > Sincerely,
> >
> > Authors

---

### Author Response · Authors · 2022-11-19
**Summary of rebuttals**

Dear Anonymous Referees,

Thank you all. Here is a summary of rebuttal:

1. Code. We thank Reviewer KfQu and Reviewer Q6i9 for the kind suggestion of submitting the code, even though our algorithm is simple and the simulation is straightforward. In the supplementary material, the zip file contains all the matlab code needed to reproduce Figure 3 (upper-left plot) (up to the simulation variations). Note that while the core algorithm only contains a few lines of matlab code, we need to resort to a solver for assignment problem for reproducing Figure 3. The C++ file is provided along the pre-compiled mex files for both windows and mac.



2. We thank Reviewer KfQu and Reviewer 7u4U for commenting on the (theoretical) quadratic dependency on k. Indeed, the dependency is linear (see Theorem 3), with a minor modification of Lemma 13. Previously, Lemma 13 used a simplified expression from Lemma 11. Thank you for this insightful comment.

3. More explanations are provided for the phenomena in Figure 1 (c.f. Remark 2), namely, why we should always keep only one non-zero element in $\textup{thres}(\cdot)$. Basically, insufficient samples lead to a poor approximation of $B^{\natural}$ using $X^T Y$. Hence, keeping more non-zero elements in $\textup{thres}(\cdot)$ will lead to a potential decrease of the inner product $\langle \textup{thres}(X^T Y), B^{\natural} \rangle$ and thus  less satisfactory performance.  For more detailed explanation, please refer to Lemma 11 (e.g., Eq. (20)) and Lemma 14.

4. Additional numerical experiments are provided to test the algorithm on MNIST; see  the results in Appendix C, which further verify the effectiveness of our algorithm. Thank Reviewer KfQu for the good suggestion.

---

> ### Author Response · Authors · 2022-12-09
> **any feedbacks for our rebuttals?**
>
> Dear Referees, Area Chair, and Senior Area Chairs,
>
> Thank you again for reviewing our paper and providing constuctive comments. As the period of interactive feedback/discussion is close to an end, we wonder if our rebuttals have satisfactoriy addressed the questions/concerns raised in the initial reviews. Also, if there are additional/new questions from the reviewers, we are happy to respond as well.  Thank you.
>
> Sincerely,
>
> Authors

---

### Decision · Program_Chairs · 2023-01-20

**Decision:**

Reject

**Justification For Why Not Higher Score:**

see above

**Justification For Why Not Lower Score:**

N/A

**Metareview: Summary, Strengths And Weaknesses:**

We carefully discussed this paper and we think the improvement from n at least quadratic in k to n at least linear in k is interesting, but ultimately decided not to base our decision on this as it seemed like a substantial change from the submitted version and potentially unfair to allow for this extra result late in the process. The fact that the stable rank has to be at least k^2 was also quite worrisome, as it required the ground-truth matrix to be extremely sparse and of limited use in practice. We think the paper would benefit from a serious revision.

**Summary Of Ac-Reviewer Meeting:**

We discussed the points above. In particular, no one had verified the n >= k modifications as the improvement from quadratic to linear seemed beyond the scope of the initial submission. Also, the stable rank assumption was unclear to us initially but we fleshed it out and it seems to have the issue pointed out above.